



# Coupling the TKE-ACM2 Planetary Boundary Layer Scheme with the Building Effect Parameterization Model

Wanliang Zhang[1], Chao Ren[2], Edward Yan Yung Ng[3], Michael Mau Fung Wong[1], and Jimmy Chi Hung Fung[1,4]

[1]Division of Environment and Sustainability, Hong Kong University of Science and Technology, Hong Kong, China
[2]Division of Landscape Architecture, Department of Architecture, Faculty of Architecture, University of Hong Kong, Hong Kong, China
[3]School of Architecture, The Chinese University of Hong Kong, Shatin NT, Hong Kong, China
[4]Department of Mathematics, Hong Kong University of Science and Technology, Hong Kong, China

**Correspondence:** Jimmy Chi Hung Fung (majfung@ust.hk)

**Abstract.** Understanding and modeling the turbulent transport of surface layer fluxes plays a critical role in a numerical weather forecasting model. The presence of heterogeneous surface obstacles (buildings) that have dimensions comparable to the model vertical resolution requires further complexity and design in the planetary boundary layer (PBL) scheme. In this study, we develop the numerical method to couple one of the recently validated PBL schemes, TKE-ACM2, with the multi-layer Building Effect Parameterization (BEP) model in WRF. Subsequently, the performance of TKE-ACM2+BEP has been examined under idealized convective atmospheric conditions with a simplified building layout. Furthermore, its reproducibility is benchmarked with one of the state-of-the-art large-eddy simulation models, PALM, which can explicitly resolve the building aerodynamics. The result indicates that TKE-ACM2+BEP outperforms the other operational PBL scheme (Boulac) coupled with BEP by reducing the bias in both the potential temperature ($\theta$) and wind speed ($u$). Following this, real case simulations are conducted for a highly urbanized domain, i.e., the Pearl River Delta (PRD) region in China. The high-resolution wind speed LiDAR observations suggest that TKE-ACM2+BEP can mitigate the overestimation in the lower part of the boundary layer compared to the Bulk method at a LiDAR site located in a densely built environment. In addition, the surface temperature and relative humidity can be improved in TKE-ACM2+BEP at surface stations in urbanized areas compared to TKE-ACM2 without BEP. However, it is revealed that BEP may not always imply a better reproduction of surface wind speed as it could exert excessive aerodynamic drag.

## 1 Introduction

Urbanization is a ubiquitous phenomenon that is widely seen across the globe. The unprecedented rate of urbanization results in more structures being constructed in populated cities, complicating the response of the incoming airflow when it encounters building clusters in the urban canopy layer (UCL) and the overlying roughness sub-layer (RSL) (Rotach, 1999). This RSL is characterized by strong turbulence due to the presence of buildings which separates the mean airflow and forms the wake region (Cleugh and Grimmond, 2012), and has significant impacts on the vertical transport of momentum and scalars over urban



regions (Roth, 2000). Parameterizations of the net sub-grid effects imposed by the building obstacles in the heavily populated cities are necessary for the mesoscale numerical weather prediction models given their horizontal resolution is typically 10-50 times larger than the street canyon scale (Britter and Hanna, 2003) which is unable to resolve the aerodynamics explicitly. In the widely-used Weather Research and Forecasting (WRF) model (Skamarock et al., 2019), the surface shear stress exerted by any type of ground obstacle can be simply parameterized using the well-known Monin-Obukhov similarity theory (MOST) by defining a friction velocity, $u_*$, which is known as the "Bulk" scheme (Liu et al., 2006). Studies have been carried out to determine different roughness lengths, $z_0$, a prerequisite for $u_*$, to account for the heterogeneity of land type (Davenport et al., 2000). However, the Bulk scheme has certain limitations, such as poorly represented urban geometry and inadequacy of applying MOST when applied to the whole RSL (Rotach, 1993), albeit it is commonly used for real-time weather forecasts (Liu et al., 2006) and the effects of built-up lands on land-see breeze circulations (Lo et al., 2007).

The single-layer urban canopy model (SLUCM) (Kusaka et al., 2001; Kusaka and Kimura, 2004) is a moderately complex urban parameterization scheme in WRF that considers the exchange of momentum and energy between the three-dimensional urban surfaces with the atmosphere in the idealized infinitely long street canyons. One remarkable drawback of SLUCM is that only the first model layer experiences the momentum and sensible heat fluxes ($F_i$ for $i \in \{1\}$, where $i$ is the vertical index at the model center) due to the presence of buildings, which could lead to unrealistically predicted prognostic variables in the upper surface layer over medium- to high-rise building cluster regions, such as the Pearl River Delta (PRD) region in southern China. In contrast, the multi-layer urban canopy models, Building Effects Parameterizations (Martilli et al., 2002, BEP), and BEP coupled to the Building Energy Model (Salamanca and Martilli, 2010, BEP+BEM), are of a higher hierarchy in urban effects parameterizations because of their capabilities in recognizing the vertically varied interactions between the atmosphere and buildings (Chen et al., 2011), i.e., $F_i$ for $i \in \{1, I_{UCL}\}$, where $I_{UCL}$ is the maximum probable vertical index within UCL. Besides the direct impact of buildings on the atmosphere dynamics and thermodynamics, modifications to two length scales in the dissipation term of the prognostic turbulent kinetic energy (TKE) equation are offered by BEP/ BEP+BEM (Martilli et al., 2002) to account for the altered vortexes' size. Studies reveal that meteorological fields and urban heat island circulation can be better reproduced utilizing the BEP/ BEP+BEM models worldwide, such as in Hong Kong (Wang et al., 2017), Barcelona (Ribeiro et al., 2021), and Bolzano (Pappaccogli et al., 2021).

However, multi-layer BEP/BEP+BEM models are not practiced as ubiquitously as the Bulk scheme or the SLUCM because they work with few planetary boundary layer (PBL) schemes [e.g., Boulac (Bougeault and Lacarrere, 1989), MYJ (Janjić, 1994), and YSU (Hong et al., 2006) added by Hendricks et al. (2020) recently]. Considering particular PBL schemes may be preferable for different regional and seasonal simulations (García-Díez et al., 2013), there is a need to couple BEP/ BEP+BEM with other WRF PBL schemes (Martilli et al., 2009).

PBL schemes that redistribute the surface fluxes and calculate the vertical mixing have profound effects on accurately depicting the meteorological conditions (Xie and Fung, 2014; Wang and Hu, 2021). A number of comparative studies have been carried out demonstrating the superiority of non-local PBL schemes over local ones at convective times where the uprising plume size is comparable to the vertical grid resolution (Arregocés et al., 2021; Banks et al., 2016; Hu et al., 2010; Xie et al., 2012; Xie and Fung, 2014). With increasing affordability in increased CPU time, recent studies deploying higher-order





turbulence closure models have shown substantial improvements in wind speeds and temperature in complex atmospheric conditions compared to the first-order ones (Chen et al., 2022; Olson et al., 2019; Zonato et al., 2022).

The TKE-ACM2 PBL scheme (Zhang et al., 2024) is one of the recently developed 1.5-order schemes featuring a non-local transport component based on the transilient matrix approach adopted from Pleim (2007a, b). Zhang et al. (2024) has shown TKE-ACM2 exhibits improvements in predicting the vertical profiles of wind speeds compared to two other operational PBL schemes [Boulac and ACM2 (Pleim, 2007b)], however, overestimated wind speeds persist in the entire surface layer at the urban station likely due to the discrepancy caused by the Bulk parameterization of surface layer fluxes. Therefore, this paper aims at further improving the skills of TKE-ACM2 in the urbanized area through:

1. formulating the numerical method to couple the TKE-ACM2 PBL scheme with the multi-layer BEP model

2. validating the coupled models in a simplified building layout scenario under different idealized initial and bottom boundary conditions by benchmarking against a finer-scale and building-resolving computational fluid dynamics model, e.g., the large-eddy simulations (LES)

3. applying the coupled models to real case simulations over a densely built area, such as the PRD region, where the land occupied by medium- to high-rise buildings account for a great proportion in the total urbanized area. Subsequently, the performance of TKE-ACM2 coupled with BEP will be evaluated, with particular interest in the simulated wind speeds using the measurements from the high-resolution wind speed LiDAR

Section 2 outlines the description of the model development, the introduction of the LES tool used to validate TKE-ACM2+BEP at idealized urban morphology setup, and the information about the Local Climate Zones (LCZ) used for real case simulations. Section 3 compares the performance of both TKE-ACM2 and Boulac PBL schemes coupled to BEP with LES under different idealized convective conditions over a simplified staggered buildings layout with the Bulk methods being the references. Section 4 presents the sensitivity of wind speed profiles to UCM and the results of real case simulations using TKE-ACM2 and Boulac PBL schemes over a month in the year 2022.

## 2    Methodology and materials

### 2.1    Numerical method to couple TKE-ACM2 and BEP

The formulation and validation of the TKE-ACM2 PBL scheme are detailed in Zhang et al. (2024). The remarkable difference between TKE-ACM2 and its predecessor ACM2 (Pleim, 2007b) lies in that TKE-ACM2 adopts the 1.5-order turbulence closure model to calculate the eddy diffusivity/viscosity, rather than using prescribed profiles for different stabilities. Moreover, TKE-ACM2 differs from Boulac in the way that the non-local transport of both the momentum and scalars under convective conditions are reflected using the transilient matrix approach in TKE-ACM2, whereas Boulac parameterizes the transport of momentum based on the local gradient only and uses the counter-gradient method for potential temperature transport which is not energy conservative. Following Pleim (2007b), the governing equation balancing the tendency terms for zonal ($u$) or

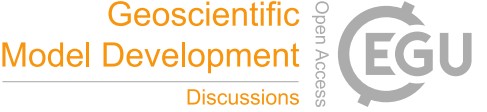

meridional ($v$) wind, potential temperature ($\theta$), and water vapor mixing ratio ($q$) with the vertical gradients of fluxes is written as,

$$\frac{\partial \zeta}{\partial t} = -\frac{\partial}{\partial z}\overline{w'\zeta'} \tag{1}$$

where $\zeta \in \{u, v, \theta, q\}$, and the vertical turbulent fluxes consisting of the local gradient transport and transilient non-local transport are parameterized as,

$$\overline{w'\zeta'}_I = -K_I \frac{S_I (\zeta_{i+1} - \zeta_i)}{V_i \Delta z_I} + \mathrm{Mu}\,(h - z_I)\,(\zeta_1 - \zeta_i) \tag{2}$$

where subscripts $i$ ($I$) denote variables located at half (full) sigma levels, $K$ is the eddy viscosity/diffusivity, $V$ and $S$ are the volume and surface fraction not occupied by buildings, $\mathrm{Mu}$ is the upward convective transport rate, and $h$ is the boundary layer height. Adding the environmental forcing acting on multiple model levels, the discretized form of Eqn.1 can then be written as,

$$\frac{\zeta_i^{n+1} - \zeta_i^n}{\Delta t} = \underbrace{f_{\mathrm{conv}}\,\mathrm{Mu}\zeta_1}_{\text{upward convective transport}} \underbrace{- f_{\mathrm{conv}}\,\mathrm{Md}_i\zeta_i + f_{\mathrm{conv}}\,\mathrm{Md}_{i+1}\zeta_{i+1}\frac{\Delta z_{i+1}}{\Delta z_i}}_{\text{donward transport}}$$
$$+ \underbrace{(1 - f_{\mathrm{conv}})\frac{1}{V_i \Delta z_i}\left[ S_I \frac{K_I (\zeta_{i+1} - \zeta_i)}{\Delta z_I} - S_{I-1}\frac{K_{I-1}(\zeta_i - \zeta_{i-1})}{\Delta z_{I-1}}\right]}_{\text{local transport}} + \underbrace{\frac{F_i}{\Delta z_i}}_{\text{env. forcing}} \tag{3}$$

where superscript $n+1$ means the time forwarding by a time-step ($\Delta t$), $f_{\mathrm{conv}}$ is the ratio that partitions the local and non-local transport, $\mathrm{Md}$ is the downward compensatory transport rate, and $\Delta z$ is the vertical resolution. The original formulation of $f_{\mathrm{conv}}$, $\mathrm{Mu}$, and $\mathrm{Md}$ are detailed in Pleim (2007b), and the model sensitivity to some of these parameters can be found in Zhang et al. (2024). The last term on the RHS of Eqn.3 denoting the collective forcing from both the urban area and natural area for the first model layer ($i = 1$) is computed using the weighted sum approach if the urban fraction (Urb) is less than 1, i.e., $F_1 = (1 - \mathrm{Urb})F_{natural,1} + \mathrm{Urb}F_{urban,1}$. For model layers $i > 1$ the environmental forcing $F_i$ is $\mathrm{Urb}F_{urban,i}$ alone. Readers interested in the parameterization of $F_{urban}$ are referred to Martilli et al. (2002). Effectively, $F_i$ is computed in the subroutine phys/module_sf_bep.F, resulting in the term $F_i/\Delta z_i$ written as the combination of implicit ($A_i$) and explicit ($B_i$) parts, i.e., $F_i/\Delta z_i = A_i\zeta_i + B_i$ for matrix inversion.

The prognostic equation of TKE ($e$) in TKE-ACM2 coupled with BEP remains identical to Zhang et al. (2024), but the parameterizations of each source/sink term are modified mainly to account for 1) external TKE source converted from mean kinetic energy when flow separates, and 2) altered characteristic length scale for eddies in the wake region due to buildings. According to Bougeault and Lacarrere (1989) and Makedonas et al. (2021), the prognostic equation of $e$ by considering the building effects can be written as,

$$\frac{\partial e}{\partial t} = -\frac{1}{\rho}\frac{\partial}{\partial z}\rho\overline{w'e} - \overline{u'w'}\frac{\partial u}{\partial z} - \overline{v'w'}\frac{\partial v}{\partial z} + \beta\overline{w'\theta'} - \epsilon + \frac{\partial F}{\partial z} \tag{4}$$





where $\rho$ is the density of air, $\beta$ is the buoyancy coefficient, and $\epsilon = \rho C_\epsilon e^{3/2}/l_\epsilon$ represents the TKE dissipation rate with $C_\epsilon$ and $l_\epsilon$ being an empirical constant and the characteristic length of energy-containing eddies, respectively. The turbulent fluxes for momentum and heat are already given in Eqn.2. $\partial F/\partial z$ representing TKE generated by buildings can be written in a similar manner to momentum/heat as $Ae+B$ which are readily available from the BEP module in WRF. Assuming the vertical turbulent transport of TKE mimics that of the passive scalar, the parameterization of $\overline{w'e}$ is expressed similarly to Eqn.2:

$$\overline{w'e}_I = -K_{e,I}\frac{S_I(e_{i+1}-e_i)}{V_i\Delta z_I} + \mathrm{Mu}\,(h-z_I)\,(e_1-e_i) \tag{5}$$

The eddy diffusivity for scalar ($K_h$) and TKE ($K_e$) are equal in magnitude and can be related to eddy viscosity ($K_m$) using the turbulent Prandtl number ($\mathrm{Pr}_t$):

$$K_e = K_h = K_m/\mathrm{Pr}_t \tag{6}$$

where $K_m = C_K l_k e^{1/2}$, $C_K$ is a $\mathcal{O}(1)$ empirical constant, and the length scale $l_k$ is modified from that calculated in Bougeault and Lacarrere (1989) ($l_{k,\mathrm{old}}$) because the buildings can generate vortices whose size of $l_{\mathrm{build.}}$ is comparable to the building spatial dimension (typically the building height) according to Martilli et al. (2002), as shown in Eqn.7.,

$$\frac{1}{l_k} = \frac{1}{l_{\mathrm{old}}} + \frac{1}{l_{\mathrm{build.}}} \tag{7}$$

Likewise, the same modification applies to $l_\epsilon$, which effectively indicates an enhanced dissipation of TKE (Martilli et al., 2002).

With the aforementioned parameterizations of intermediate parameters, Eqn.3 can be solved by writing into a linear system of $\mathbf{Ax}=\mathbf{b}$, where the column vector $\mathbf{x}$ contains the unknown prognostic variable $\zeta_i^{n+1}$, the square boarded band matrix $\mathbf{A}$ is the coefficient matrix which consists of the first column entry ($\mathbf{E}$), diagonal ($\mathbf{D}$), upper diagonal ($\mathbf{U}$), and lower diagonal ($\mathbf{L}$) elements, and the column vector $\mathbf{b}$ is composed of the explicit terms in Eqn.3. To keep the same order of numerical accuracy as TKE-ACM2 (Zhang et al., 2024), the Crank-Nicolson scheme is retained which splits $\zeta_i^{n+1}$ to $C\zeta_i^{n+1} + (1-C)\zeta_i^n$ with $C=0.5$ being the Crank-Nicolson factor. Subsequently, the $i-$th row and $i-$th column element of $\mathbf{D}$ can be expressed as,

$$\mathbf{D}_{i,i} = 1 + C f_{\mathrm{conv}}\,\mathrm{Md}_i\Delta t$$
$$+ C\,(1-f_{\mathrm{conv}})\frac{\Delta t}{\Delta z_i}\left(\frac{K_I S_I}{\Delta z_I} + \frac{K_{I-1}S_{I-1}}{\Delta z_{I-1}}\right) - C A_i\Delta t \tag{8}$$

The $i-$th row element of column vector $\mathbf{b}$ is expressed as:

$$\mathbf{b}_i = \zeta_i^n + (1-C)f_{\mathrm{conv}}\,\mathrm{Mu}\,\zeta_1^n\Delta t$$
$$- (1-C)f_{\mathrm{conv}}\,\zeta_i^n\Delta t + (1-C)f_{\mathrm{conv}}\,\mathrm{Md}_{i+1}\zeta_I^n\frac{\Delta z_{i+1}}{\Delta z_i}\Delta t$$
$$+ \frac{1-C}{V_i\Delta z_i}f_{\mathrm{conv}}\,(K_I S_I\frac{\zeta_{i+1}^n - \zeta_i^n}{\Delta z_I}$$
$$- K_{I-1}S_{I-1}\frac{\zeta_i^n - \zeta_{i-1}^n}{\Delta z_{I-1}})\Delta t + (1-C)A_i\zeta_i^n\Delta t + B_i\Delta t \tag{9}$$

The $i-$th row and $j-$th column element of $\mathbf{U}$, $\mathbf{L}$, and $\mathbf{E}$, are the same to Eqn.13, Eqn.14, and Eqn.15 in Zhang et al. (2024), respectively, except an additional multiple of $S_I/V_i$ applies to $K_I$.





## 2.2 Large-eddy simulation model

Prior to implementing the TKE-ACM2 PBL scheme coupled with BEP in real case simulations, we performed idealized

simulations using prescribed surface heat fluxes along with simplified urban morphology and benchmarked it against one of the state-of-the-art and building-aerodynamics-resolved large-eddy simulation (LES) models, i.e., the PALM model. The PALM model (Maronga et al., 2015; Raasch and Schröter, 2001) is a non-hydrostatic incompressible Navier-Stokes equation solver, which is rigorously evaluated against experiments and thus often serves as the benchmark for deriving new parameterizations regarding the boundary layer turbulent mixing processes in the mesoscale weather forecasting model. It utilizes a 1.5-order

turbulence closure model solving anisotropic turbulence in three dimensions simultaneously. One salient advantage of PALM compared to other wall-resolved LES is that PALM adopts the MOST between the solid boundary and the first model layer above, which greatly elevates the computational efficiency while preserving the accuracy in the context that the mesoscale model has $\Delta z \in \{\mathcal{O}(10)\,\text{m}, \mathcal{O}(1000)\,\text{m}\}$.

## 2.3 Idealized simulations setup

A $1\,\text{km}$ long by $1\,\text{km}$ wide by $1.5\,\text{km}$ high domain with equidistant spatial resolution $\Delta x = \Delta y = \Delta z = 5\,\text{m}$ with staggered building arrays is set up for the PALM model. Fig.1 provides a plan view for the domain setup and urban morphology configuration, where the spatial dimension of the building follows a $20\,\text{m}$ square cross-section with a height of $40\,\text{m}$ and the windward wall faces perpendicularly to the upwind flow. The prescribed height of building arrays is justified by that it is commonly seen in the domain of interest (Hong Kong)(Kwok et al., 2020). The street width in both horizontal directions is chosen as $30\,\text{m}$ to

mimic a moderately densely built environment. Such street width to building width ratio $(2/3)$ is deemed an "open" exposure in urban areas and has good representativity in Hong Kong. Unlike PALM runs at the building-resolved scale, WRF+BEP is at the building-parameterized scale ($\Delta x = \Delta y = 1\,\text{km}$) where explicitly resolving the building aerodynamics is impractical. Therefore, we prescribe the urban morphological parameters to be consistent with PALM in the WRF+BEP look-up table required for BEP. The horizontal extension of WRF+BEP domain is set as $20\,\text{km}$ by $20\,\text{km}$ to accommodate the thermal plumes

of large scale in the convective flow (Schmidt and Schumann, 1989), while the same height of $1.5\,\text{km}$ is retained but with slightly coarser $\Delta z = 12.5\,\text{m}$.

     The initial condition is prescribed as $u_0 = u_g = 10\,\text{m/s}$ uniformly distributed along the vertical direction and $v_0 = v_g = 0\,\text{m/s}$, where $u_g$ and $v_g$ are the geostrophic winds with Coriolis parameter being $10^{-4}\,\text{s}^{-1}$. Two initial potential temperature profiles are selected for the idealized simulations, one being the moderately convective atmosphere ($\overline{w'\theta'}_0 = 0.10\,\text{K}\,\text{m}^{-1}\,\text{s}^{-1}$,

denoted as Case 10WC) with no capping inversion and the other representing strongly convective atmospheric stability ($\overline{w'\theta'}_0 = 0.24\,\text{K}\,\text{m}^{-1}\,\text{s}^{-1}$, denoted as Case 24SC) with a strong capping inversion to limit the growth of boundary layer. The analytical expressions of two initial $\theta$ profiles are listed below, where $\partial\theta/\partial z$ in the free atmospheres is $1\,\text{K}/100\,\text{m}$ in both cases,



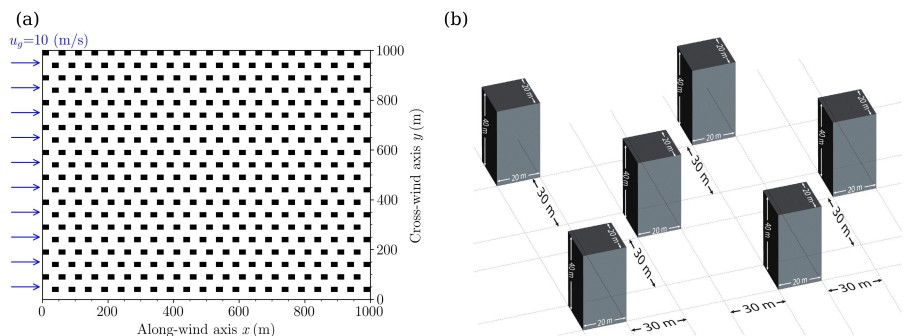

**Figure 1.** a) The plan view for domain setup; b) urban morphology configuration.

$$\text{Case 10WC}: \quad \theta_0(z_i) = \begin{cases} 300\text{K}, & z_i < 600\text{m} \\ 300\text{K} + 1/100(z_i - 600)\text{K}, & z_i \geq 600\text{m} \end{cases} \quad (10)$$

and

$$\text{Case 24SC}: \quad \theta_0(z_i) = \begin{cases} 300\,\text{K}, & z_i < 600\,\text{m} \\ 300\text{K} + 6/100(z_i - 600), & 600\,\text{m} \leq z_i < 800\,\text{m} \\ 300\,\text{K} + 6/100(800 - 600)\text{K} + 1/100(z_i - 800)\,\text{K}, & z_i \geq 800\text{m} \end{cases} \quad (11)$$

All boundary conditions are identically set in both PALM and WRF+BEP simulations. The lateral boundary conditions for the along-wind direction ($x$) and cross-wind direction ($y$) are periodic to simulate an infinitely long urban fetch. The bottom boundary conditions for heat are reflected by different values of $\overline{w'\theta'}_0$ and the top boundary condition is free-slip. The microscale roughness length ($z_0$) for both the ground and roof is chosen as $0.01\,\text{m}$ in PALM, ensuring consistency with the value in the look-up table used in WRF+BEP. The runtime parameters that are crucial to obtain meaningful results (Ayotte et al.,

1996; Nazarian et al., 2020) for PALM and WRF+BEP are described in a more detailed way in Section 3. The temperature of solid surfaces (roof, wall, and streets) in PALM and WRF+BEP are prescribed as $300\,\text{K}$.

### 2.4 Local Climate Zones (LCZ)

Default 21-class MODIS or 24-class USGS landuse dataset may be obsolete or inaccurate in capturing the high heterogeneity of the urbanized surfaces in a particular domain. For instance, only 3 urban types are distinguished in the MODIS/USGS, resulting

in a less resolved variability in urban morphological parameters in different urban landuse. In contrast, the LCZ classification scheme provides a more detailed description of urban landuse at the local scale ($10^{-2}$ to $10^{-4}\,\text{m}$). The superiority of LCZ classification over MODIS is proven in Hong Kong region revealed by Du et al. (2023). Each urban type represents a distinct urban morphology configuration, and LCZ class 1 to 10 covers from the most compactly built to the sparsely built environment,





which can greatly aid the investigation of meteorological variables in a highly heterogeneous domain. In this study, we adopted the recently updated and well-validated LCZ global map developed by Demuzere et al. (2022) to characterize the urban surfaces in the innermost nested domain (D4), shown in Fig.2b. The distribution of 17 classes is plotted in logarithmic $y$ scale in Fig.2c. What should be highlighted from the LCZ map is that the proportion of compact high-rise type (LCZ 1) and open high-rise type (LCZ 2) is considerably higher compared to European cities focusing on a similar scale, e.g., Vienna (Hammerberg et al., 2018), Barcelona (Ribeiro et al., 2021), reinforcing the demand of a multi-layer building effects parameterization coupled to the PBL scheme.

## 2.5   Real case simulations setup

A four-nested domain adopting the parent domain grid ratio of 1:3 with a reference latitude of 28.5 °N and a longitude of 114 °E (Fig.2a) is chosen. The coarsest domain (D1) with $\Delta x = \Delta y = 27\,\mathrm{km}$ spans 283 grid points in the East-West direction and 184 in the North-South direction, covering the entire China. The finest domain (D4) with a horizontal resolution of $1\,\mathrm{km}$ focuses on the PRD region, where there exist a few heavily populated and densely built mega-cities including Hong Kong (7.3), Shenzhen (17.6), and Guangzhou (18.7), with the number in the brackets showing the population in the unit of million as of the year 2021. The surface stations and high-resolution wind speed LiDAR locations deployed in D4 are highlighted in Fig.2d. 30-day simulations are performed between July 18 20 o'clock local time (LT) to August 18 20LT in 2022. The integration is performed by overlapping one day as the spin-up between two consecutive four-day segments.

We configured WRF eta levels such that multiple vertical model grids can intersect the UCL and RSL. Thus, the variability of wind speeds can be better represented by BEP in the presence of buildings taller than the first above-ground full eta level. The lowest 6 half eta levels correspond to approximately $9\,\mathrm{m}$, $28\,\mathrm{m}$, $49\,\mathrm{m}$, $71\,\mathrm{m}$, $96\,\mathrm{m}$, and $122\,\mathrm{m}$ above ground level (AGL). We used NCEP GFS analysis data at 6-hourly input intervals to provide the initial and lateral boundary conditions. Noah land-surface model and revised MM5 surface layer scheme have been selected. Four simulations were performed using different PBL schemes and UCM. Namely, they are TKE-ACM2+Bulk, TKE-ACM2+BEP, Boulac+Bulk, and Boulac+BEP. Other relevant setup in the namelist is detailed in Table A1. Meanwhile, the look-up table for LCZ class properties which provides crucial parameters including impervious fraction and building height distribution is also attached in the supplementary Zhang (2024).

## 3   Idealized simulations results

Nazarian et al. (2020) show the importance of choosing appropriate runtime parameters for LES in a neutral atmosphere over building arrays. Since this study adopts two convective scenarios in a similar urbanized domain, extra attention must be paid to the thermal characteristics in determining the runtime parameters. As revealed by Ayotte et al. (1996) and Shin and Dudhia (2016), the duration of simulations can be determined by examining the temporal variation of turbulence statistics. We first examined the time required for LES to reach a quasi-equilibrium state by investigating the variation of the maximum resolved TKE ($e_{\mathrm{res.}}$) and the absolute value of maximum vertical velocity ($|w_{\mathrm{max}}|$), shown in Fig.3.



**Figure 2.** a) The four-nested domain; b) LCZ classification for 1-km resolution domain 4 with the color scheme represented in panel c); c) LCZs distribution in D4; d) Distribution of surface stations and wind speed LiDAR.

It is found the quasi-equilibrium is reached in both cases when LES runs for approximately $10.2\tau$, where $\tau = h/w^*$ is the convective time scale and $w^* = (\beta\overline{w'\theta'}_0 h)^{1/3}$ is the convective velocity scale. The 10.2 large-eddy turnover time is deemed a reasonable indicator for well-developed dynamic fields over the domain with buildings compared to other studies adopting a factor of 5 (Ayotte et al., 1996; Pleim, 2007b; Zhang et al., 2024) and 6 (Shin and Dudhia, 2016) over a flat domain. The $u, \theta$ fields are horizontally averaged at $10.2\tau$ and used as initial conditions to drive WRF+BEP simulations for another $20\tau$.

Subsequently, the results during the last $6\tau$ corresponding to $3600\,\mathrm{s}$ or $2400\,\mathrm{s}$ are horizontally and temporally averaged for





**Figure 3.** a) The time series of $e_{\mathrm{res.}}$ for Case 10WC; b) The time series of $|w_{\mathrm{max}}|$ for Case 10WC; c) Same to a) but for Case 24SC; d) Same to b) but for Case 24SC.

comparison. The $6\tau$ averaging interval is consistent with Ayotte et al. (1996); Pleim (2007b); Zhang et al. (2024). Table 1 summarizes the key turbulence characteristics of the convective flow and the runtime parameters.





**Table 1.** Turbulence characteristics and runtime parameters

| Parameter | Case 10WC | Case 24SC |
|---|---|---|
| Capping inversion strength | N.A. | $\frac{\partial \theta}{\partial z} = 6/100$, K/m |
| PBL height, $h(t = 10.2\tau)$ | 840 m | 720 m |
| Large-eddy turnover time, $\tau$ | 600 s | 404 s |
| Convective velocity scale, $w^*$ | 1.40 m/s | 1.78 m/s |
| Spin-up time $(10.2\tau)$ | 6,300 s | 4,200 s |
| Duration of simulation $(30\tau)$ | 18,000 s | 12,000 s |
| Averaging time (last $6\tau$) | 3,600 s | 2,400 s |

The horizontally averaged $u, \theta$ profiles during the last $6\tau$ are displayed in Fig.4. Meanwhile, the turbulence fluxes outputted from PALM and computed from WRF PBL schemes are plotted in Fig.5. The total turbulent momentum flux in Boulac is simply

computed using the local gradient as $-K_m \partial u / \partial z$ while the turbulent heat flux needs to add the counter-gradient flux shown as $-K_h(\partial \theta / \partial z - \gamma)$. The root-mean-square-error (RMSE) for first-order moments $u, \theta$ and second-order moments $\overline{w'\theta'}, \overline{w'u'}$ calculated below the PBL height is displayed in Fig.6.

In Case 10WC where there is a moderate surface heat flux, both TKE-ACM2+BEP and Boulac+BEP can reproduce the unstable atmosphere below the inertial sub-layer (ISL) which is located at approximately $3H$. However, Boulac+BEP predicts

a warmer bias in $\theta$ at the first model layer and a colder bias at the roof level, resulting in an excessively unstable UCL. In addition, the cold bias persists in the entire mixed layer. On the contrary, TKE-ACM2+BEP produces a less warm bias in the UCL. Furthermore, $\theta$ in the overlaying ISL is well reproduced by TKE-ACM2+BEP. A deeper well-mixed boundary layer $(\partial \theta / \partial z \approx 0)$ can be simulated by TKE-ACM2+BEP, while discrepancy from LES results is discovered in Boulac+BEP which suggests the boundary layer becomes stable from approximately $10H$. In comparison to BEP simulations, Bulk methods

produce consistently overestimated $\theta$ within the PBL. Inspecting the heat flux profile in Fig.5a reveals that the trends of variation are well captured in the two BEP simulations where the sink of heat flux is reproduced when approaching the roof level from the ground. This is attributed to the prescribed temperature of the building wall and roof (300 K) being cooler than the atmosphere, leading to conduction. A possible explanation for the warm bias observed in both Bulk simulations is the lack of heat sink beyond the first model layer. In general, TKE-ACM2+BEP simulates a better matched $\overline{w'\theta'}$ profile in the mixed

layer. Boulac+BEP produces $\overline{w'\theta'}$ vertical profile of a weaker magnitude, which may account for the $\theta$ profile becoming stable from $10H$. However, greater discrepancies are observed in TKE-ACM2+BEP within UCL and near PBL height where the relatively constant $\overline{w'\theta'}$ in the middle UCL is not exhibited in either BEP simulation. The inflection point in the wind speed profile is well simulated at the roof level in BEP and PALM, opposite to Bulk simulations where the wind shear $(\partial u / \partial z)$ is relatively gentle at the roof level. The momentum simulated by TKE-ACM2+BEP generally provides a better reproduction

than Boulac+BEP, particularly in the mixed layer. The most visible negative bias of $u$ in BEP simulations occurs at $[1H, 5H]$. It should be highlighted that from the ground level up to the top of UCL, both BEP simulations result in an overestimated wind

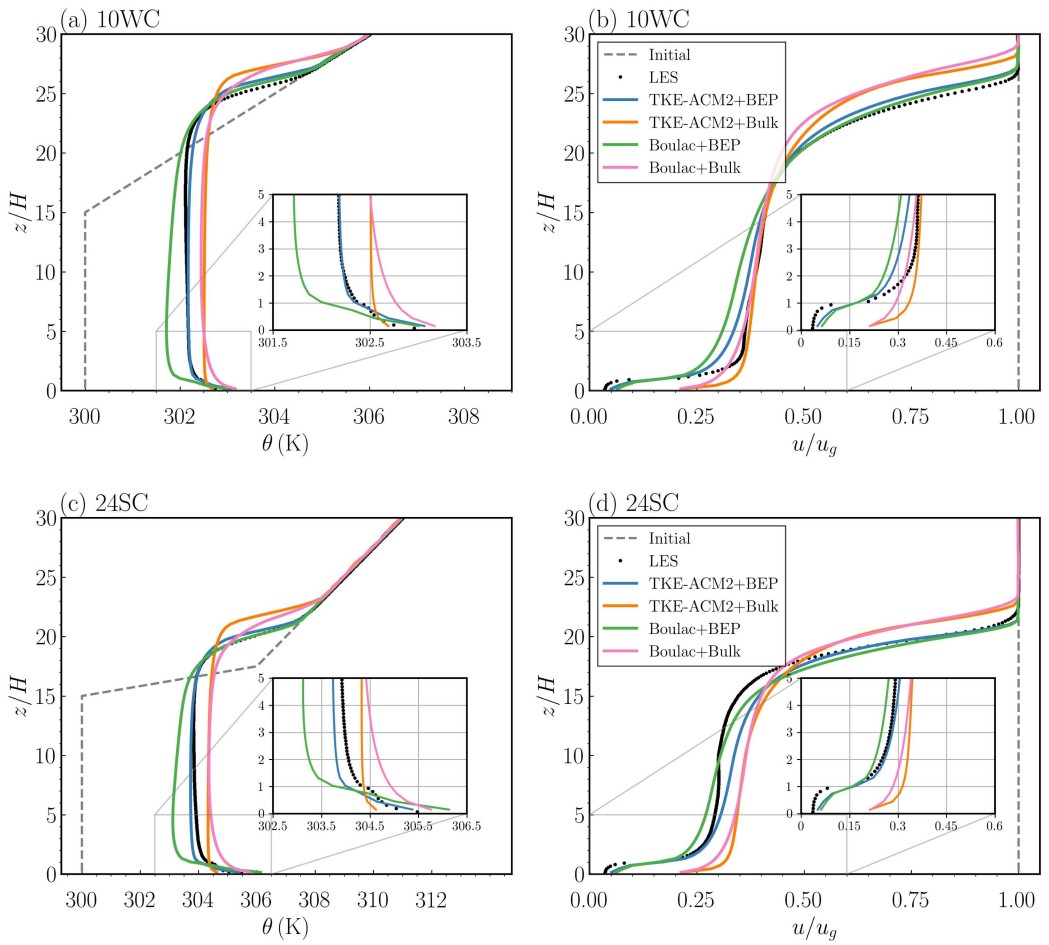

**Figure 4.** a) Horizontally averaged $\theta$ profile during the last $6\tau$ for Case 10WC; b) Horizontally averaged $u$ profile normalized by $u_g = 10\,\mathrm{m/s}$ during the last $6\tau$ for Case 10WC; c) Same to a) but for Case 24SC; d) Same to b) but for Case 24SC. The grey dashed lines represent the initial conditions. The black dots denote the LES results. The solid blue, orange, green, and pink lines represent TKE-ACM2+BEP, TKE-ACM2+Bulk, Boulac+BEP, and Boulac+Bulk, respectively.

speed as opposed to an underestimation in the mixed layer. This has shown the wind shear at the roof level is underestimated if the buildings are parameterized following BEP in contrast to that they are explicitly resolved. The Bulk simulations clearly indicate the lack of multi-layer parameterization of aerodynamic drag leads to significantly overestimated wind speed within the UCL. It is discovered that the momentum flux increases from zero at the ground level to a maximum value at some height followed by a descending trend in BEP simulations, in contrast to the monotonically descending trend in simulations where the Bulk method is adopted. However, the magnitude of $\overline{w'u'}$ simulated by the two schemes exhibit greater discrepancies than that of $\overline{w'\theta'}$. Boulac+BEP produces consistently underestimated $\overline{w'u'}$. TKE-ACM2+BEP results in a slightly less biased $\overline{w'u'}$



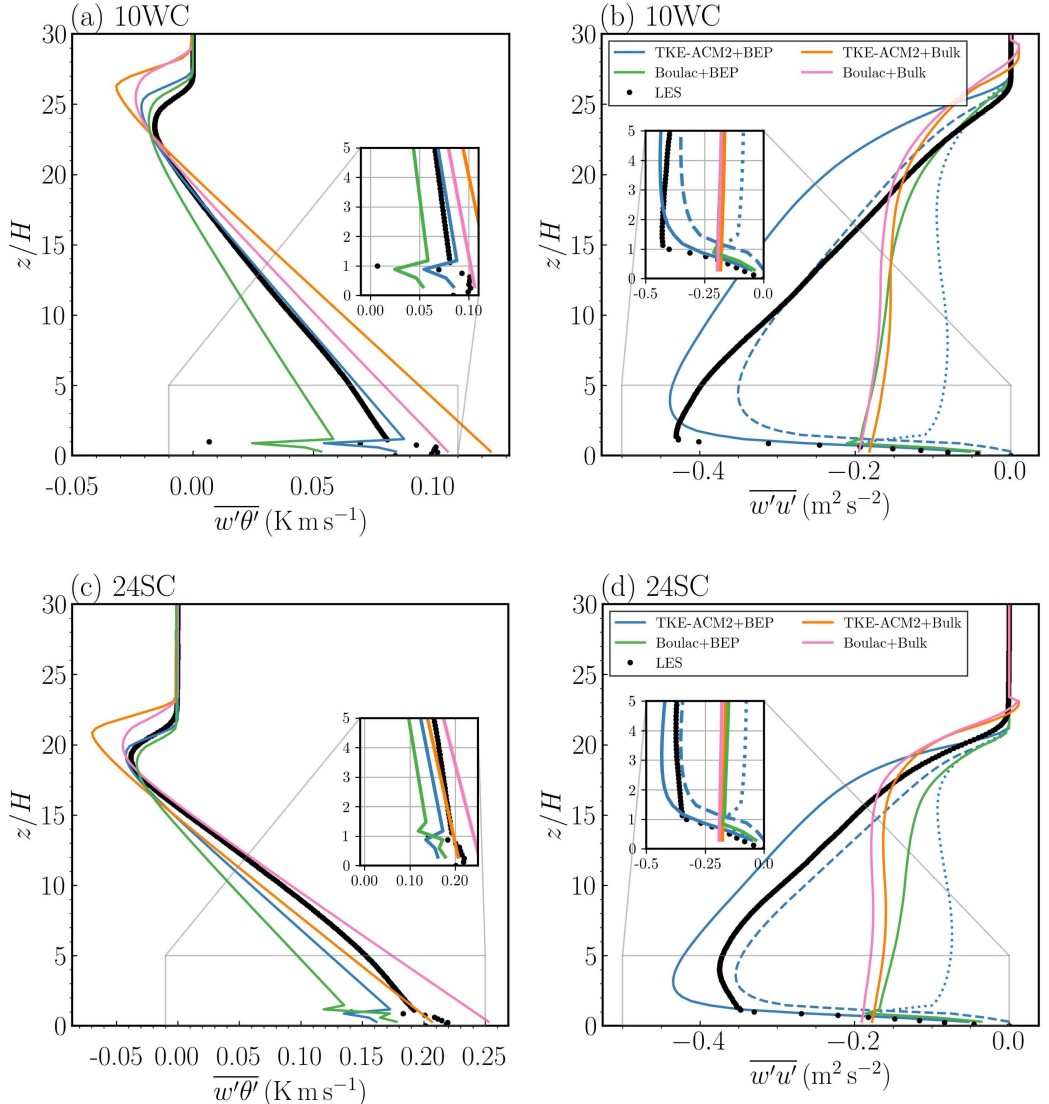

**Figure 5.** a) Horizontally averaged $\overline{w'\theta'}$ profile during the last $6\tau$ for Case 10WC; b) Horizontally averaged $\overline{w'u'}$ profile during the last $6\tau$ for Case 10WC; c) Same to a) but for Case 24SC; d) Same to b) but for Case 24SC. The black dots denote the LES results. The solid blue, orange, green, and pink lines represent TKE-ACM2+BEP, TKE-ACM2+Bulk, Boulac+BEP, and Boulac+Bulk, respectively. The blue dashed and blue dotted lines represent the local and non-local turbulent momentum fluxes of TKE-ACM2+BEP, respectively. The profiles are zoomed in aside for $z/H \leq 5$. All $y-$ axes are normalized by the uniform building height $H = 40\,\mathrm{m}$. The profiles are zoomed in aside for $z/H \leq 5$. All $y-$axes are normalized by the uniform building height $H = 40\,\mathrm{m}$.

but the maximum value occurs at approximately $z/H = 4$, whereas LES suggests the height at which $\overline{w'u'}$ peaks, $z/H_{\overline{w'u'}_{\max}}$, 
is 1 in this case. The closer match of $\overline{w'u'}$ simulated by TKE-ACM2+BEP is attributed to the non-local momentum flux



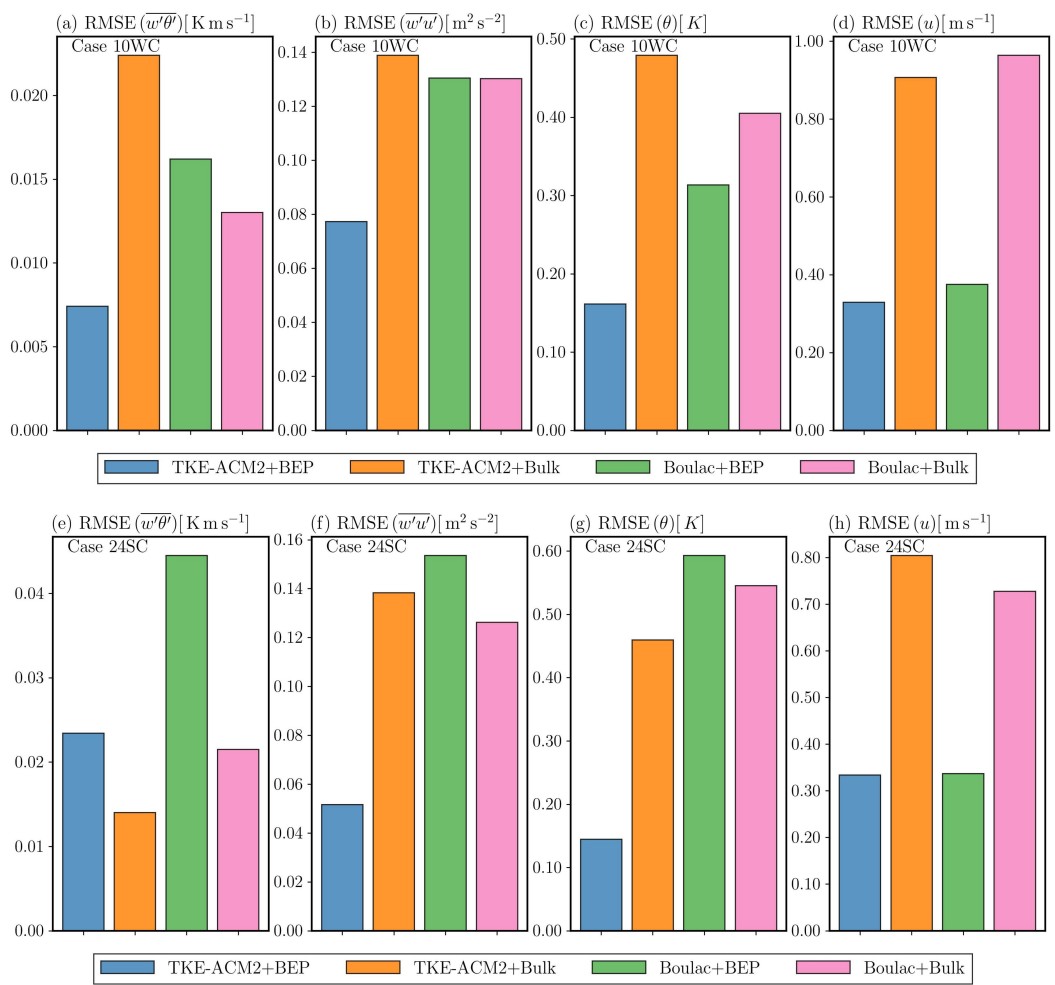

**Figure 6.** a-d): RMSE for $\overline{w'\theta'}$, $\overline{w'u'}$, $\theta$, and $u$ calculated below the PBL height for Case 10WC, respectively; e-h): Same as a-d) but for Case 24SC.

contributing to a considerable proprortion. In summary, TKE-ACM2+BEP is able to simulate a well-mixed boundary layer under such a prescribed convective atmospheric stability. The inflection point at the roof level can be reproduced similarly to how Boulac+BEP behaves. In addition, TKE-ACM2+BEP exhibits better ability in predicting the $\theta$ and $u$ profile which is reflected in $\sim$48.5% reduction of RMSE($\theta$) and $\sim$12.2% reduction in RMSE($u$) compared to Boulac+BEP.

Similar behaviors of the two schemes are found in Case 24SC, where TKE-ACM2+BEP simulates a notably less warm bias in the UCL, particularly at the first model layer. Additionally, the $\theta$ profile extending from the UCL up to $18H$ is considerably better reproduced by TKE-ACM2+BEP whereas Boulac+BEP predicts consistently cold bias below the inversion. This can be likely attributed to the more underestimated $\overline{w'\theta'}$ in the mixed layer simulated by Boulac+BEP (Fig.5c). The two Bulk simulations persist the warm bias throughout the PBL as in Case 10WC. A few similarities have been found in the momentum profile





between the 24SC and 10WC cases. Firstly, Boulac+BEP predicts consistently lower wind speed than TKE-ACM2+BEP. Secondly, both BEP simulations tend to yield overestimated wind speed in the UCL. Thirdly, Bulk methods cannot reproduce the inflection point and exhibit the greatest positive bias in the UCL. Lastly, the wind shear at the roof level displays lower magnitudes in the two BEP simulations compared to LES. The difference in performance is that TKE-ACM2+BEP exhibits slightly less deviation from $1H$ to approximately $5H$, but starts to visibly overpredict in $[7H, 17H]$. In contrast, Boulac+BEP

shows negative bias in $[1H, 7H]$ and provides a promising match in $[7H, 17H]$. Fig.5d indicates TKE-ACM2+BEP can generate $\overline{w'u'}$ of a similar pattern and magnitude to LES in the whole PBL, whereas Boulac+BEP seems to largely underestimate the momentum flux as observed in the 10WC case. A notable difference in $z/H_{\overline{w'u'}_{\max}}$ is found between 24SC and 10WC: LES shows $z/H_{\overline{w'u'}_{\max}}$ increases from $z/H = 1$ to approximately $z/H = 4$ when $\overline{w'u'}$ is stronger. Further analysis by partitioning the total $\overline{w'u'}$ reveals that the non-local component plays a more important role in distributing the surface layer fluxes

to the mixed layer in TKE-ACM2+BEP, as reflected by the red dashed line in Fig.5d. Case 24SC suggests TKE-ACM2+BEP provides a closer match in the magnitude and shape heat flux profile compared to Boulac+BEP when $\overline{w'u'}_0$ increases. Also, TKE-ACM2+BEP reports $z/H_{\overline{w'u'}_{\max}} = 3$, which deviates less than Boulac+BEP ($z/H_{\overline{w'u'}_{\max}} = 1$), compared to LES results. Rotach (2001) investigated several field measurements and wind tunnel experiments to examine the height of the maximum turbulence momentum flux. Their results show that $\overline{w'u'}$ can occur at approximately $3H$ which is deemed as the top of the ISL.

This indicates that stronger heat flux can cause elevated $z/H_{\overline{w'u'}_{\max}}$, requiring extra caution in the PBL scheme when dealing with a sizable urban morphology. Conclusively, TKE-ACM2+BEP outperforms Boulac+BEP in Case 24SC in simulating the $\theta$ profile by reducing RMSE by 75.6%, which is consistent with the closer match of $\overline{w'u'}$ in the mixed layer. The RMSE($u$) is fount to be both $0.33\,\mathrm{m\,s^{-1}}$ in TKE-ACM2+BEP and Boulac+BEP. As a result, the performance of the two PBL schemes coupled with BEP in predicting the momentum profiles is comparable statistically below the PBL height and exhibits considerable

superiority over the Bulk methods.

## 4  Real case simulations results

### 4.1  Effects of BEP on $U$ and $\theta$ over different LCZ types

Fig.A1 and Fig.A2 display the vertical profiles of $\theta$ and $U = \sqrt{u^2 + v^2}$ averaged in the entire simulated month over LCZ classes 1-10, water surfaces, and other natural landuse. What should be highlighted from Fig.A1 and Fig.A2 is that TKE-ACM2+BEP

produces larger $U$ and warmer $\theta$ within the PBL height compared to Boulac+BEP over all urban grids, corroborating the findings in Section 3. The differences in vertical profiles of $U$ and $\theta$ between the BEP and Bulk methods are shown in Fig.7 and Fig.8 with the shadowed area representing $\pm 1$ standard deviation ($\sigma$) over the whole month.

The trends of $\Delta U(\mathrm{BEP} - \mathrm{Bulk})$ are found similar in the two schemes. The height at which $\Delta U = 0$ is generally observed at $\sim 300\,\mathrm{m}$, corroborating what has been observed in Berlin, Munich, and Prague by Karlický et al. (2018). Below $z(\Delta U =$

$0)$, BEP consistently reduces $U$ by $\sim 1 - 2\,\mathrm{m/s}$, and $\Delta U(z)$ generally exhibits a monotonically descending trend from the ground level. Nonetheless, TKE-ACM2+BEP shows slightly lower $z(\Delta U = 0)$. In addition, it shows a consistently greater reduction in $\Delta U$ at the height close to the ground than Boulac+BEP in all urban grids. Another distinction is that TKE-



**Figure 7.** a-j): Monthly mean vertical profiles of $\Delta U$ over LCZ 1-10; k): over water surfaces; l): over other natural landuse. The shadowed area denotes $\pm 1\sigma$ variability.

ACM2+BEP seems to accelerate $U$ slightly more than Boulac+BEP from $z(\Delta U = 0)$ to approximately $800\,\mathrm{m}$. Both BEP simulations produce less profound differences in $U$ over the water surfaces and the natural land cover than over urban grids. Fig.8 displays the variation of $\Delta\theta(\mathrm{BEP} - \mathrm{Bulk})$, which is found to increase from negative values at the ground level to nearly zero beyond $z(\Delta\theta(\mathrm{BEP} - \mathrm{Bulk})$ in TKE-ACM2+BEP but shows a persistently negative value in Boulac+BEP. In the absence of anthropogenic heat modeled in BEP+BEM, the buildings behave as a sink of heat in the lower part of PBL by solely applying BEP.



**Figure 8.** a-j): Monthly mean vertical profiles of $\Delta\theta$ over LCZ 1-10; k): over water surfaces; l): over other natural landuse. The shadowed area denotes $\pm 1\sigma$ variability.

## 4.2 Monthly mean diurnal profiles of $U$ compared with high-resolution LiDAR measurements

Firstly, the monthly mean diurnal variation of heat flux, Monin-Obkuhov length ($L$), and the stability parameter ($h/L$) are displayed in Fig.9. The heat flux pattern does not exhibit a visible difference at the rural LiDAR station HT. At the LCZ 5 USTSS LiDAR location, introducing BEP consistently exerts a greater surface heat flux than Bulk methods. In contrast, the magnitude of surface heat flux at the LCZ 1 KP LiDAR location has been reduced in the entire diurnal cycle. The surface



**Figure 9.** Column a) plots the monthly mean diurnal pattern of the surface heat flux at USTSS, HT, and KP in the first, second, and third row, respectively; Column b) the Monin-Obkuhov length ($L$) in semi-log $y-$axis; Column c) the stability parameter ($h/L$).

station located in the proximity of the KP LiDAR site suggests that the observed $T_2$ (shown in Fig.A51) is constantly higher than the prescribed building and street temperature, which can justify that the reduction of modeled surface heat flux is caused by the sink of buildings when BEP is activated.

The wind speed LiDAR offers hourly measurements of wind speed at an altitude of $50\,\mathrm{m}$ above ground level (AGL), with vertical increments of $25\,\mathrm{m}$. The measured and simulated wind speed profiles are averaged during the whole month for each hour and are displayed in Fig.10 (USTSS), Fig.11 (HT), and Fig.12 (KP). To quantify the performance of each simulation, the RMSE and mean bias (MB) are demonstrated in Fig.13.

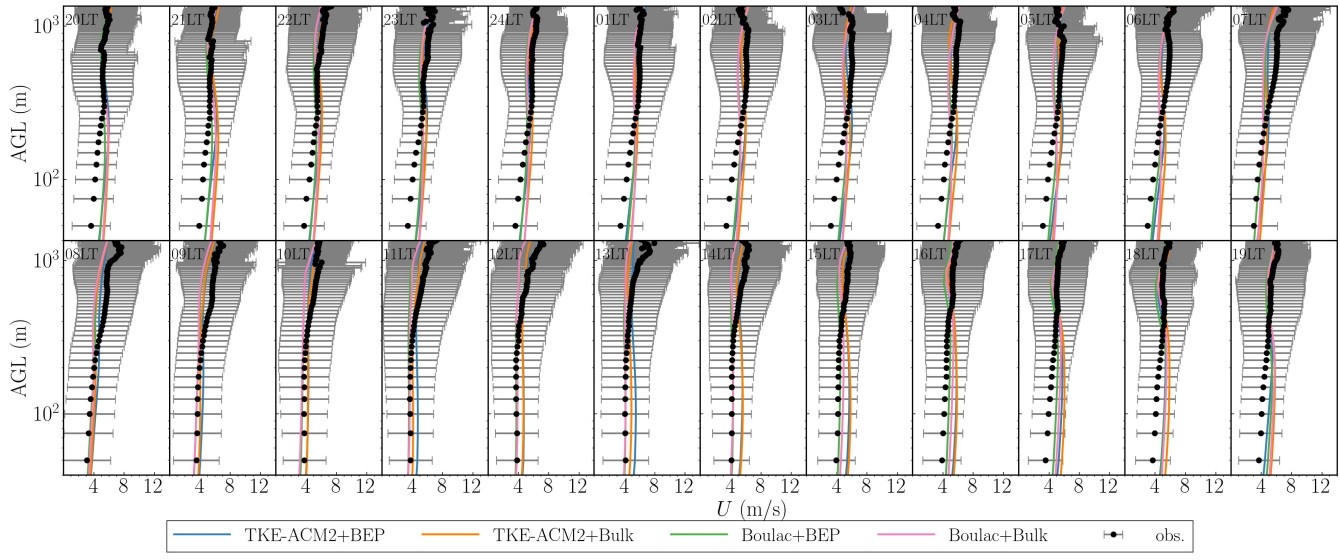

**Figure 10.** Monthly mean vertical profiles of wind speeds at USTSS LiDAR station. The blue, orange, green, and pink lines denote TKE-ACM2+BEP, TKE-ACM2+Bulk, Boulac+BEP, and Boulac+Bulk, respectively. The black dots represent the measurements from LiDAR with the error bar denoting the $\pm 1$ standard deviation.

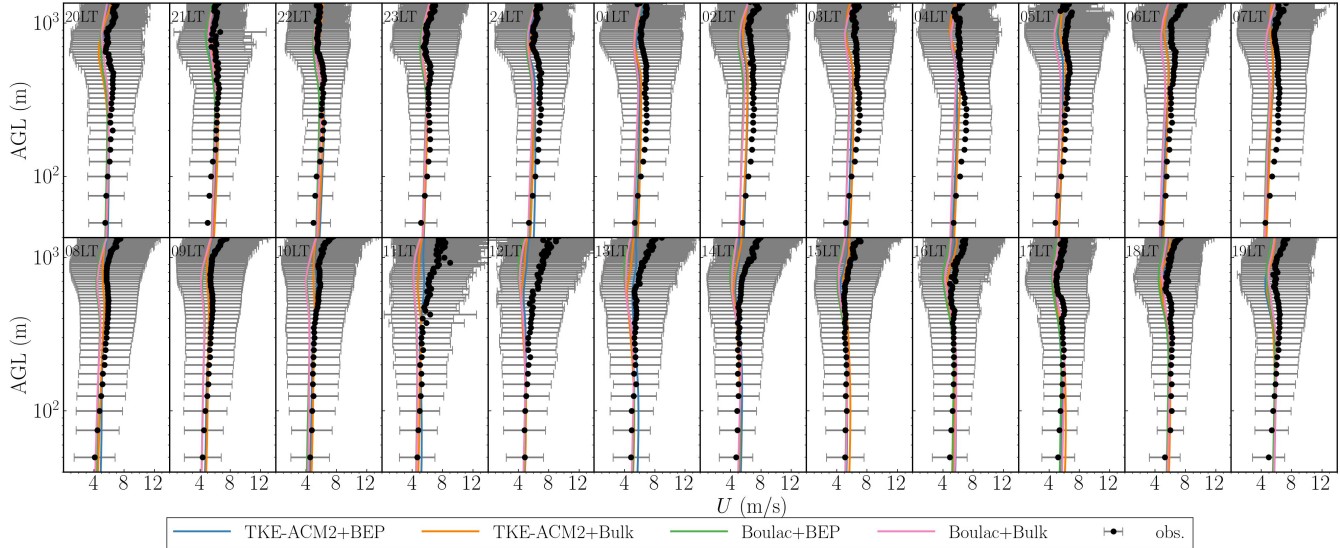

**Figure 11.** Same as Fig.10 but at HT LiDAR station.

Despite the fact that USTSS is located in a model grid identified as LCZ 5 (open midrise), applying BEP does not imply a consistent and noticeable reduction in the wind speed, which is inconsistent with an average reduction of $\sim 1 - 2\,\mathrm{m/s}$ found in all LCZ 5 grids as shown in Fig.7e. A closer inspection reveals that $\Delta U(\mathrm{BEP} - \mathrm{Bulk})$ is found to be less visible in both

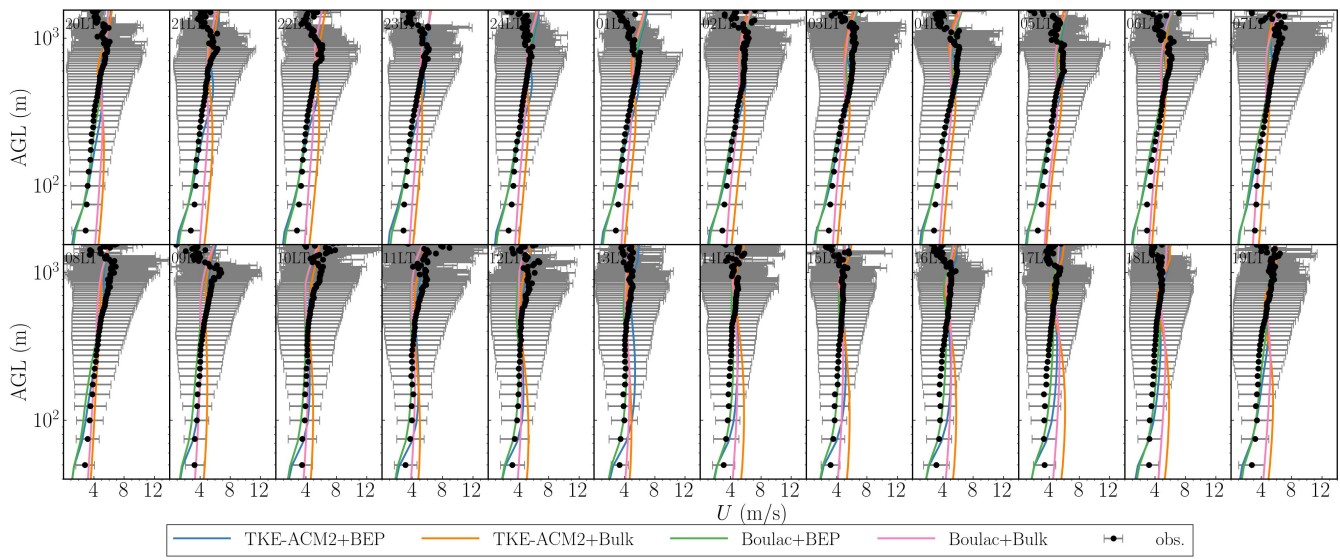

**Figure 12.** Same as Fig.10 but at KP LiDAR station.

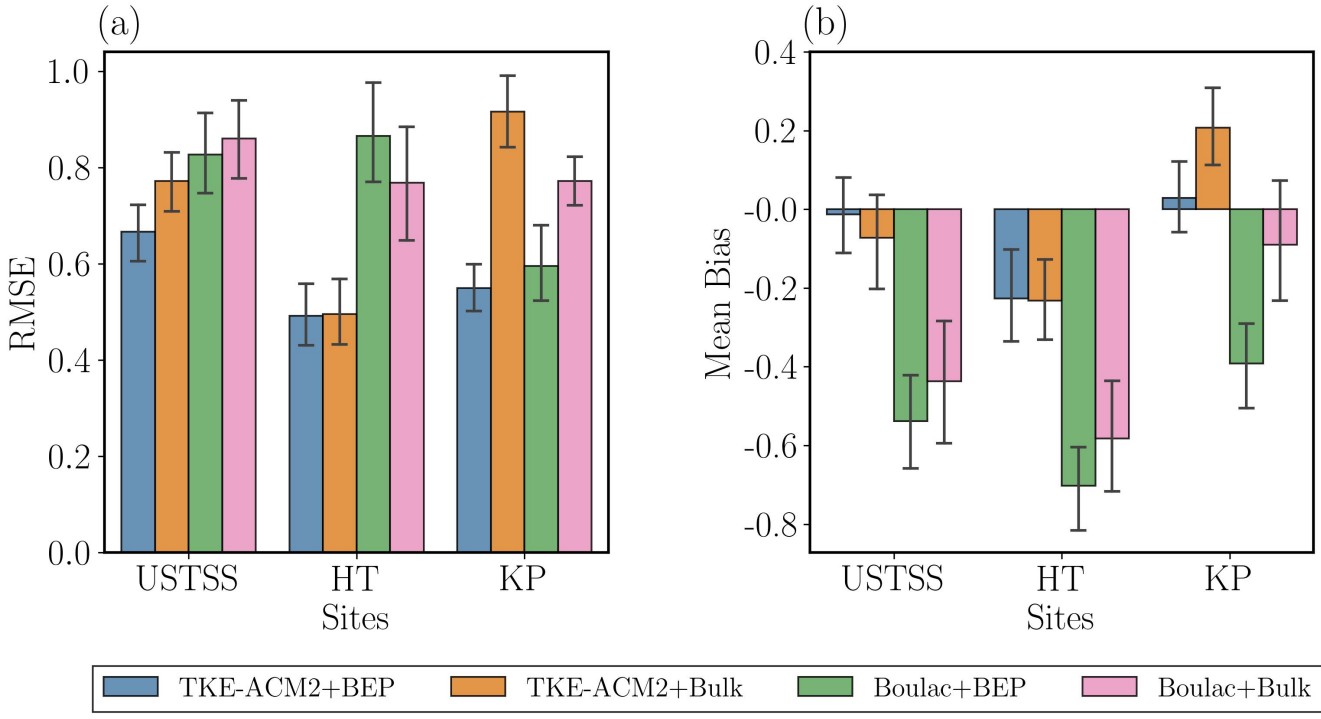

**Figure 13.** RMSE (a) and MB (b) calculated at the three LiDAR stations for four simulations.





PBL schemes compared to that observed at another urban LiDAR station KP (LCZ 1). A possible explanation is that the
LCZ map in Fig.2b indicates that the model grid containing USTSS is bordered by either natural or water grids, effectively
isolating it from other urban grids. Consequently, the wind approaching this grid experiences a less rough fetch, leading
to a reduced drag exerted on this model grid. It is found that the overprediction occurs primarily below $\sim 300\,\mathrm{m}$ during
the nighttime for all schemes, with BEP simulations producing a slightly smaller positive bias. The overestimation below
$\sim 300\,\mathrm{m}$ persists in TKE-ACM2+BEP during 11LT-17LT, whereas Boulac+BEP aligns better. Furthermore, all schemes exhibit
underestimation above $\sim 300$m. Corroborating the accelerated $\Delta U$ above $z(\Delta U = 0)$ as shown in Fig.7, TKE-ACM2+BEP
produces gently larger $U$ beyond $\sim 300$m, which leads to the least positive bias compared to LiDAR observations. It also
should be highlighted that the accelerated $U$ observed in the upper PBL is also manifested in the 10WC and 24SC idealized
cases (Fig.4b&d). Lastly, a detailed investigation reveals that the wind speed profiles during the nighttime show less difference
between each scheme compared to $U$ simulated during the daytime. This is likely due to TKE-ACM2 and Boulac adopting
a similar turbulence closure model, and their performance may differ less when there is an absence of convective thermal. In
summary, the RMSE histograms in Fig.13 show that TKE-ACM2+BEP produces the least RMSE and the least negative MB,
in contrast to Boulac+BEP further worsened the negative MB compared to Boulac+Bulk.

At the rural LiDAR station HT, the application of BEP has a limited impact on the PBL performance over non-urban
model grids, aligning with the conclusion drawn in Section 44.1. The differences between BEP and Bulk are indistinguishable
below $\sim 400\,\mathrm{m}$. However, TKE-ACM2+BEP occasionally accelerates $U$ notably beyond $\sim 600\,\mathrm{m}$, for instance, from 10LT
to 13LT, matching closer to LiDAR observations. Conclusively, BEP causes minor differences in $U$ profiles at non-urban
model grids, particularly in the lower PBL. Therefore, the differences in the predictability of $U$ within this height range over
non-urban grids are largely caused by the PBL schemes rather than the UCMs. Nonetheless, the influence of BEP could be
slightly more profound by accelerating $U$ in the upper PBL in TKE-ACM2+BEP, which leads to an improved reproduction
of $U$ profiles at HT LiDAR station. Unlike the USTSS station located in an isolated LCZ 5 grid, KP is deployed in the well-
developed downtown area of Hong Kong. The two schemes coupled with BEP exhibit considerably decelerated wind speeds
below $\sim 400\,\mathrm{m}$, collaborating Fig.7a. It should be highlighted that discrepancies are diminished mainly from $100 - 400\,\mathrm{m}$
where Bulk methods predict visibly overestimated wind speeds. However, BEP tends to over-reduce the wind speeds in both
schemes from 50 to $100\,\mathrm{m}$ which corresponds to approximately $2.5 - 5H$ for this LCZ type, particularly at heights closer to the
ground level. In contrast, Bulk methods produce bias of similar magnitudes but with a reverse sign below $100\,\mathrm{m}$. From $600 -$
$1000\,\mathrm{m}$, Boulac+BEP exhibits the lowest wind speeds during the daytime and is comparable with TKE-ACM2+BEP during
the nighttime. Moreover, the performance of TKE-ACM2+BEP differs from that of Boulac+BEP between $600 - 1000\,\mathrm{m}$ by
generating a less negatively biased wind speed at particular hours, e.g., from 08LT to 12LT. Holistically, the two PBL schemes
coupled with BEP lead to considerably improved RMSE compared to the Bulk methods at this particular compact high-rise grid.
More specifically, TKE-ACM2+BEP significantly outperforms TKE-ACM2+Bulk, reducing RMSE from $0.92\,\mathrm{m/s}$ to $0.55\,\mathrm{m/s}$
and reducing MB from $0.21\,\mathrm{m/s}$ to $0.03\,\mathrm{m/s}$. Additionally, TKE-ACM2+BEP demonstrates slightly better performance than
Boulac+BEP of which RMSE is $0.60\,\mathrm{m/s}$.





### 4.3 Effects of BEP on $U_{10}$, $T_2$, and $\mathrm{RH}_2$ over different LCZ types

The diurnal patterns of $U_{10}$, $T_2$, and $2\,\mathrm{m}$ relative humidity ($\mathrm{RH}_2$) simulated by the four schemes are demonstrated in Fig.A3,
Fig.A4, and Fig.A5, respectively. Fig.14, Fig.15, and Fig.16 display the differences between BEP and Bulk simulations in
monthly averaged $U_{10}$, $T_2$, and $\mathrm{RH}_2$ by grouping into different LCZ types. It is found that BEP generally reduces $U_{10}$ in
urban grids. Boulac suggests $\Delta U_{10}$ follows an evident diurnal pattern while TKE-ACM2 shows a less profound one with a
different phase. In general, $\Delta U_{10}$ simulated by Boulac reaches the minimum absolute value at approximately 10LT. In this
case, $|\Delta U_{10}|_{\min}$ can be as low as zero. In contrast, $|\Delta U_{10}|_{\min}$ is more likely to be found at around 06LT in TKE-ACM2, where
its magnitude is larger than Boulac. The diurnal pattern simulated by Boulac persists in the natural grids, and accelerated
$U_{10}$ are observed during the daytime. $\Delta T_2$ and $\Delta \mathrm{RH}_2$ exhibit evident diurnal variation which are negatively correlated due
to the inverse proportionality. Unlike the $\Delta U_{10}$ diurnal pattern, similar phases in $\Delta T_2$ and $\Delta \mathrm{RH}_2$ are found between the two
schemes. It is noticeable that Boulac+BEP may occasionally generate a warmer $T_2$ than Boulac+Bulk, e.g., at ∼11 LT to
12LT over LCZ 2, 4, and 5 grids, but TKE-ACM2+BEP consistently reduces $T_2$. Correspondingly, a drier atmosphere could
be observed in Boulac+BEP than Boulac+Bulk, but $\mathrm{RH}_2$ always increases in TKE-ACM2+BEP. Lastly, the three surface
meteorological variables exhibit little sensitivity to surface layer fluxes parameterizations over water grids but can be gently
altered at other natural grids.

### 4.4 Monthly mean diurnal patterns of $U_{10}$, $T_2$, and $\mathrm{RH}_2$ compared with surface stations

The time series and metrics for each individual station are provided in the supplementary Zhang (2024) for detailed visu-
alization. The diurnal variations of $U_{10}$, $T_2$, and $\mathrm{RH}_2$ for a total of 31 surface stations are aggregated based on their LCZ
classifications, which are shown in Fig.17, Fig.18, and Fig.19, respectively. The RMSE histograms are displayed in Fig.20.
Applying BEP results in a significant reduction in $U_{10}$, which is consistent with the trend observed in Fig.14 for all LCZ urban
grids. This reduction greatly improves the predictions of $U_{10}$ at LCZ 5, 6, and 8 stations, which are landuse consisting of
primarily low- or mid-rise buildings at relatively low building density. A closer inspection shows that the improvements are
more profound during the nighttime over the aforementioned stations. Among these stations, TKE-ACM2+BEP performs the
best or comparably to Boulac+BEP by reaching an RMSE as low as $1.0\,\mathrm{m/s}$. However, the modeled wind speeds in BEP for
LCZ 1 (compact high-rise), 4 (open high-rise), and 10 (heavy industry) stations are undesirably lower than the observed values,
particularly during the daytime. The largely underestimated $U_{10}$ at LCZ 1 surface station is consistent with the underestimation
of $U$ at $25\,\mathrm{m}$ observed at the KP LiDAR station. More specifically, the two BEP simulations produce $U_{10} \approx 1\,\mathrm{m/s}$ constantly
and exhibit an RMSE $1.7 - 2.4\,\mathrm{m/s}$, which is consistently worse than that of Bulk methods (RMSE∼ $1.5\,\mathrm{m/s}$). The excessive
reduction in $U_{10}$ is likely to be caused by the mismatched local LCZ class ($100\,\mathrm{m}$ resolution) and re-gridded LCZ class ($1\,\mathrm{km}$
resolution) at LCZ 1, 4, and 10 stations, which is also reported in (Ribeiro et al., 2021). For instance, the KP surface station
which is identified as an LCZ 1 class station by WRF is deployed at the hill whose spatial scale is $50\,\mathrm{m}$. Therefore, the exposure
is relatively open and flat in the local vicinity of the KP surface station. As a consequence, the model simulated $U_{10}$ is largely
underpredicted and not representative enough at the exact location of the station. Lastly, it is observed that BEP coupled with



**Figure 14.** a-j): Monthly mean $\Delta U_{10}$ over LCZ 1-10; k): over water surfaces; l): over other natural landuse. The shadowed area denotes $\pm 1\sigma$ variability.

the two schemes causes subtle differences to $U_{10}$ at non-urban stations, but the slightly decreased wind speeds match better with observations at stations located on water surfaces by reducing RMSE by $\sim 0.2 \text{m/s}$.

Coinciding with Fig.15, $T_2$ at nighttime has been reduced in BEP simulations over all LCZ urban stations. The change in $T_2$ at daytime is less profound, and Boulac+BEP is likely to produce a warmer daytime $T_2$ compared to TKE-ACM2+BEP. As
a result, both PBL schemes coupled with BEP considerably improve the warm bias at nighttime compared to Bulk methods, and their predictability at daytime changes insignificantly. TKE-ACM2+BEP behaves visibly better than Boulac+BEP over





**Figure 15.** Same as Fig.14 but for $T_2$.

LCZ 2, 5, 8, and 10 stations, where the RMSE($T_2$) is reduced by $0.51\,\mathrm{K}$, $0.13\,\mathrm{K}$, $0.27\,\mathrm{K}$, and $0.11\,\mathrm{K}$, respectively. Otherwise, their performance over other LCZ urban stations is comparable. The four simulations generate $T_2$ diurnal cycles of much less amplitude than observations at water surfaces, where inter-scheme difference is marginal and each scheme deviates by $\sim 2.0\,\mathrm{K}$

from observations. $T_2$ at rural stations is found to be constantly underestimated by all simulations, especially the reduction of nighttime $T_2$ by BEP simulations tends to slightly worsen the underestimation.

The trend of $\mathrm{RH}_2$ is inversely proportional to that of $T_2$. It is found that Boulac always generates a much dryer boundary layer at all types of surface stations, regardless of the choice of surface layer flux parameterizations. Likewise to Boulac+Bulk,



**Figure 16.** Same as Fig.14 but for $RH_2$.

TKE-ACM2-Bulk produces a dryer surface layer though of much less negative bias. The addition of BEP to the two PBL schemes has greatly improved the predictability of $RH_2$ at urban stations, while the influence of BEP is relatively marginal on $RH_2$ at non-urban stations. Fig.16 indicates that BEP implies a greater $RH_2$ when coupled to TKE-ACM2 than to Boulac, resulting in a more profound improvement in TKE-ACM2+BEP. In summary, BEP does not only affect the surface wind speed but also has important implications on the $T_2$ and $RH_2$ diurnal patterns. TKE-ACM2+BEP has demonstrated superiority over other schemes in reducing the warm bias at nighttime and enhancing the predictability of $RH_2$ at urban stations.



**Figure 17.** Comparison of monthly mean diurnal patterns of $U_{10}$ with surface stations. The title describes the LCZ type and the associated number of surface stations. The blue, orange, green, and pink lines represent the TKE-ACM2+BEP, TKE-ACM+Bulk, Boulac+BEP, and Boulac+Bulk, respectively. The black marker denotes the surface station observations. The shadowed area displays $\pm 1\sigma$ variability.





**Figure 18.** Same as Fig.17 but for $T_2$ comparison.

## 5 Conclusions

In this study, we have developed the numerical method to couple BEP with the TKE-ACM2 planetary boundary layer scheme detailed in Zhang et al. (2024). We first evaluated the performance of TKE-ACM2+BEP under a series of idealized atmospheric



**Figure 19.** Same as Fig.17 but for $RH_2$ comparison.

conditions with a simplified urban morphology in WRF. The state-of-the-art large-eddy simulation tool, PALM, configured with three-dimensional equidistant resolution is utilized to provide a reference result at the building-resolved scale. It has been

demonstrated that TKE-ACM2+BEP significantly improves the reproduction of the vertical profiles of $\theta$ and $u$ in the two prescribed surface heat flux cases compared to TKE-ACM2 without any urban canopy model, and it also shows superiority



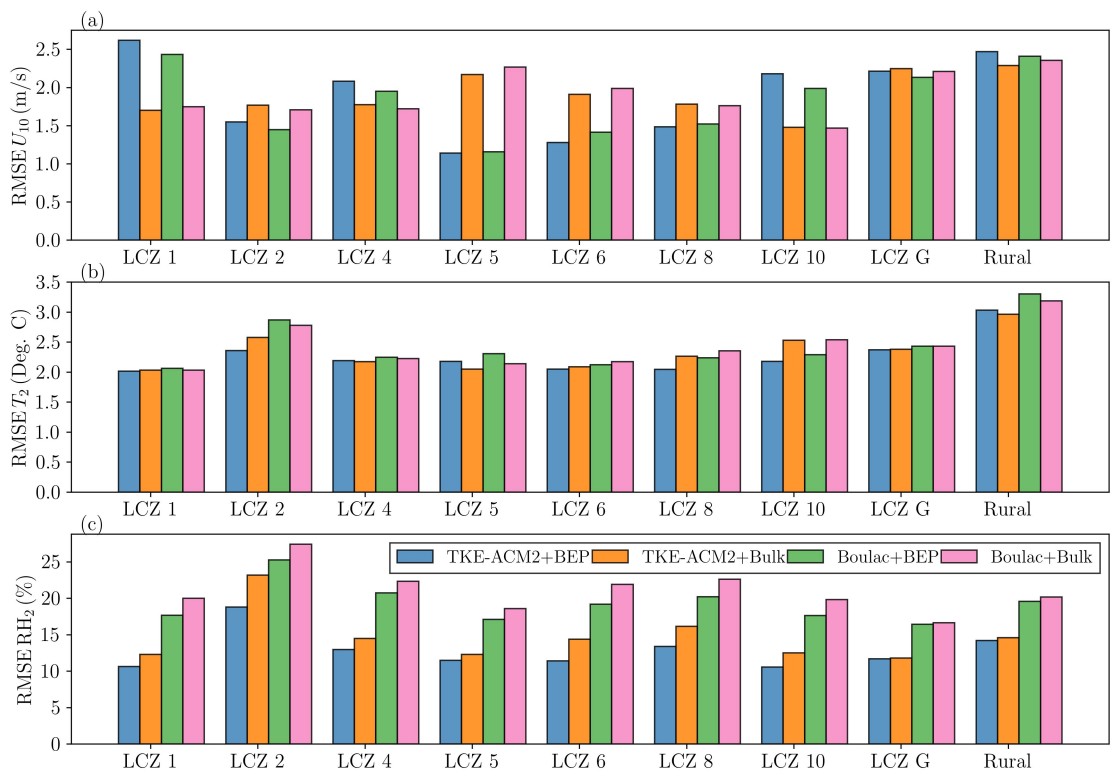

**Figure 20.** RMSE at aggregated station types, with a), b), and c) drawing for $U_{10}$, $T_2$, and $RH_2$, respectively.

in the moderately convective case compared to another widely used Boulac+BEP scheme. In particular, TKE-ACM2+BEP predicts $\theta$ with a reduced warm bias within the urban canopy layer. Additionally, Boulac+BEP produces a sharper $\partial\theta/\partial z$ at the roof level, leading to a noticeable cold bias in the mixed layer. A closer inspection suggests that turbulent fluxes are

better reproduced by TKE-ACM2+BEP, which is attributable to the non-local fluxes. In contrast, Boulac+BEP significantly underestimates their magnitudes.

Real case simulations adopting different surface layer fluxes parameterization schemes were performed. It is shown that TKE-ACM2+BEP exhibits similar behavior to Boulac+BEP, both of which reduce $U$ below a certain height over the LCZ urban grids. Likewise, BEP indicates the buildings act as a sink of heat in the lower part of the boundary layer during the simulated

period where observed $T_2$ is generally higher than the prescribed building and ground temperature. High-resolution wind speed LiDAR observations are obtained and used to evaluate the performance of TKE-ACM2+BEP. It is revealed that the reduction of $U$ at USTSS (LCZ 5) is not constantly visible across a diurnal cycle, which is likely attributed to the fact that USTSS is





located in an isolated urban grid with smoother fetch in all directions. BEP has little impact on the wind speed profiles at the rural LiDAR station HT, where the four simulations perform similarly. Lastly, TKE-ACM2+BEP demonstrates a considerably

improved performance in predicting vertical profiles of $U$ at the LCZ 1 LiDAR station compared to TKE-ACM2+Bulk. The overestimation in the lower boundary layer has been much improved. However, the wind speeds are over-reduced by BEP in Boulac and TKE-ACM2 below $\sim 100\,\mathrm{m}$. Overall, TKE-ACM2+BEP outperforms others in simulating the wind speed profile at this highly urbanized grid. It should be pointed out that BEP does not necessarily improve the prediction of $U_{10}$ at all types of urban stations as it can lead to largely underestimated $U_{10}$ compared to the two schemes with Bulk methods, for

instance, extremely low wind speeds at observed at LCZ 1, 2, 4, and 10 stations. The enhanced predictability of $U_{10}$ simulated by TKE-ACM2+BEP is noticeable at stations located in relatively low building density, such as LCZ 5, 6, and 8 stations. The non-linear feedback to $U_{10}$ at rural stations can be slightly improved by TKE-ACM2+BEP, where RMSE is reduced by $\sim 0.2\mathrm{m/s}$. Therefore, it is critical to select an appropriate configuration for simulating the wind speed in the whole boundary layer. Nonetheless, BEP has consistently improved the reproduction of $T_2$ for TKE-ACM2 over urban stations, particularly

reducing the warm bias at nighttime. The predicted $T_2$ by TKE-ACM2+BEP is generally comparable with or slightly better than Boulac+BEP at most urban stations. On the other hand, $\mathrm{RH}_2$ exhibits comparable sensitivity to PBL schemes and UCMs. BEP leads to a more moist PBL, and TKE-ACM2+BEP exhibits the least dry bias in reproducing $\mathrm{RH}_2$ among all simulations. This work does not aim to demonstrate that the new TKE-ACM2+BEP performs definitively better in simulating all aspects of the meteorological variables than other combinations of PBL and UCM; rather, it offers valuable insights for selecting

appropriate model configurations to meet various objectives regarding different atmospheric processes.





*Code and data availability.* The PALM model is an open-source atmospheric LES model under the GNU General Public License (v3). (available at https://palm.muk.uni-hannover.de/trac, last access: November 2024). The WRF model encompassing the current version of TKE-ACM2 PBL scheme used to produce the results in this paper is archived on Zenodo (Zhang, 2024) under the Creative Commons Attribution 4.0 International license, as the data simulated by PALM and WRF for idealized and real simulations, LiDAR observations, and surface station observations (Zhang, 2024).

## Appendix A

**Table A1.** Configurations of WRF version 4.3.3 settings for simulations using Boulac and TKE-ACM2 PBL schemes and UCM schemes.

| WRF version 4.3.3 Options | Settings |
| --- | --- |
| Meteorological data for boundary and initial conditions | NCEP GFS 0.25° by 0.25° latitudinal and longitudinal resolution with 6-hour interval |
| Grid resolutions | 27 km for D1 with 1:3 parent domain grid ratio for nested domains |
| Time steps | 120 s for D1 with 1:3 parent time step ratio for nested domains |
| Number of grid points (East-West × North-South) | D1 283 × 184, D2 223 × 163, D3 172 × 130, and D4 214 × 163 |
| Number of vertical eta levels | 39 |
| Pressure at top model level | 50 hPa corresponding to approximately 20 km AGL |
| Number of vertical levels in WRF Preprocessing System (WPS) output | 34 |
| Number of soil levels in WPS output | 4 |
| Microphysics scheme | WSM 3-class simple ice scheme Hong et al. (2004) |
| Longwave radiation scheme | RRTMG scheme Iacono et al. (2008) |
| Shortwave radiation scheme | RRTMG scheme Iacono et al. (2008) |
| Surface layer scheme | Revised MM5 Monin-Obukhov scheme Jiménez et al. (2012) |
| Land-surface scheme | Unified Noah land-surface model Chen and Dudhia (2001) |
| Cumulus scheme | Grell-Freitas ensemble scheme Gall et al. (2013) |
| Urban model (sf_urban_physics) | BEP (option 2) and Bulk (option 0) |
| Land-use data | LCZ (use_wudapt_lcz=1, num_land_cat=41) |
| Grid nudging | 6-hour interval grid analysis nudging only for D1 |
| Observational nudging | Off for all domains |



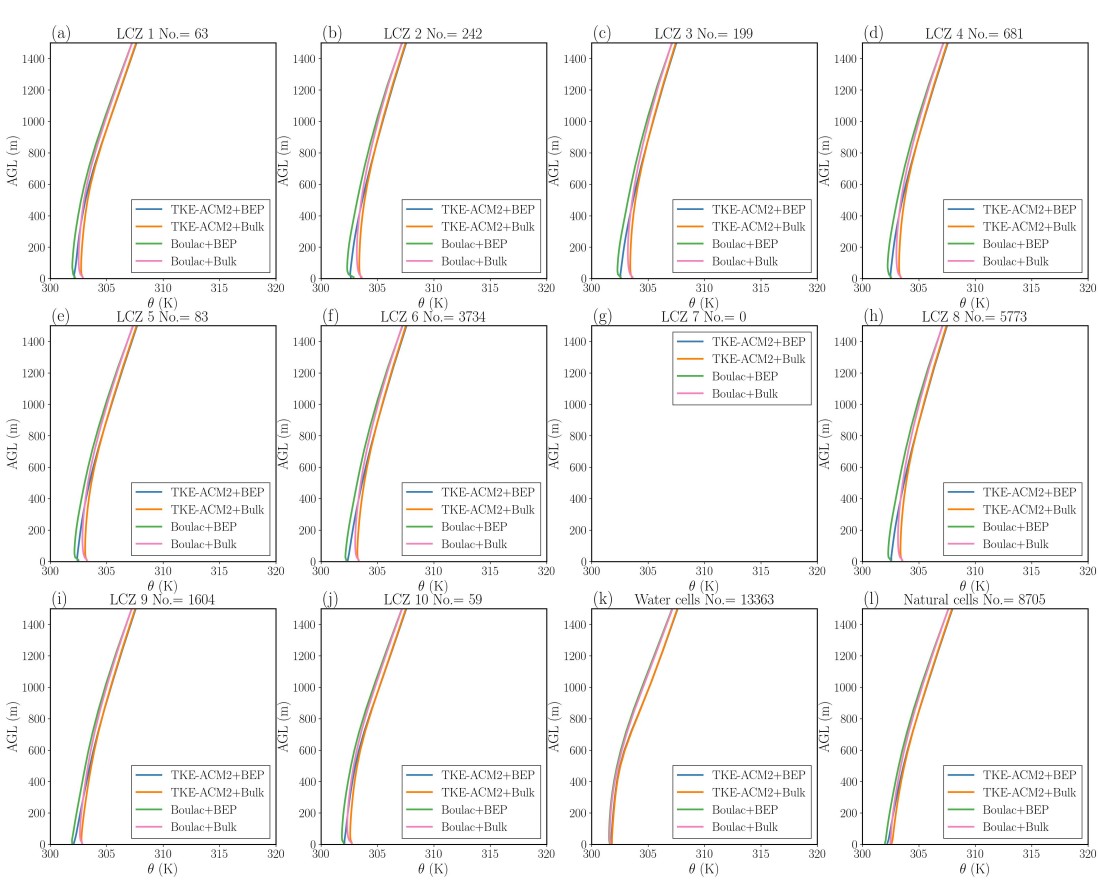

**Figure A1.** a-j): Monthly mean vertical profiles of $\theta$ over LCZ 1-10; k): over water surfaces; l): over other natural landuse.



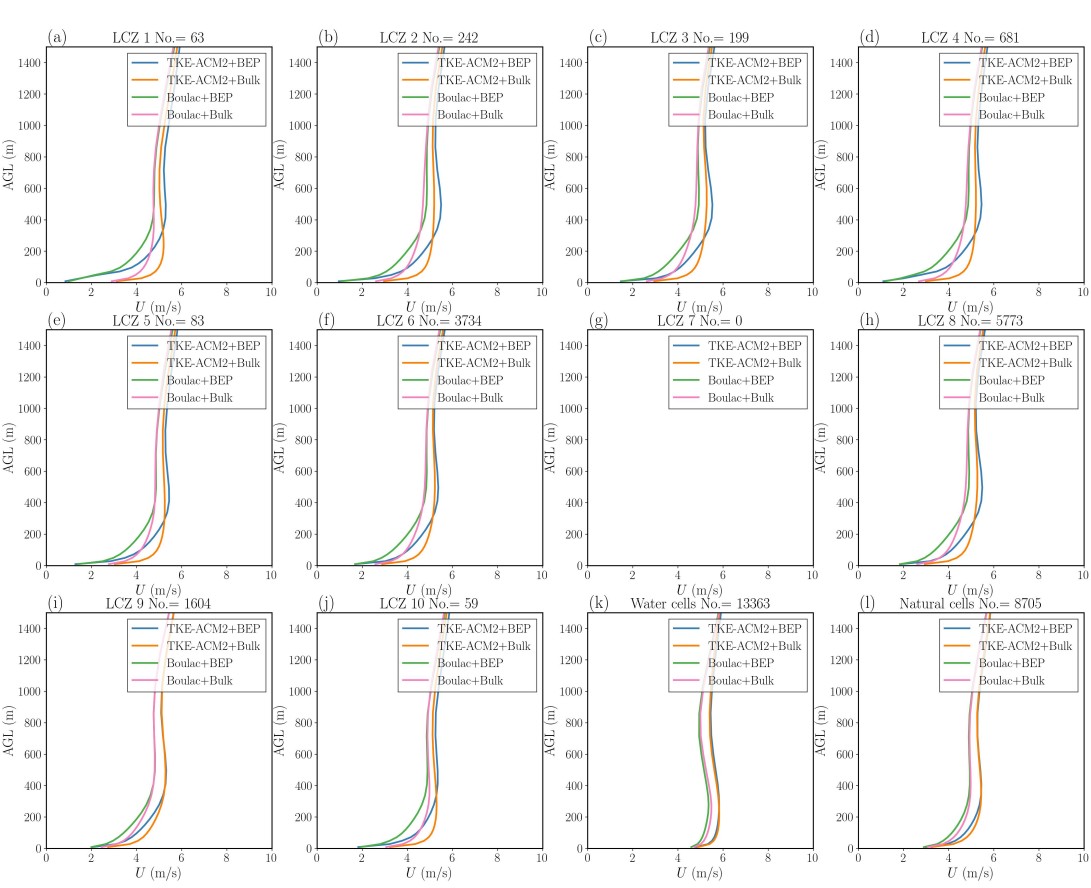

**Figure A2.** Same to Fig.A1 but for $U$.



**Figure A3.** a-j): Monthly mean $U_{10}$ over LCZ 1-10; k): over water surfaces; l): over other natural landuse.





**Figure A4.** Same to Fig.A3 but for $T_2$.



**Figure A5.** Same to Fig.A3 but for $RH_2$.





*Author contributions.* WZ: Conceptualization, methodology, data curation, validation, investigation, and formal analysis; RC and EYYN: Funding acquisition and project administration; MMFW: Methodology, data curation, and investigation; JCHF: Conceptualization, investigation, supervision, project administration, and funding acquisition. All authors reviewed and edited the manuscript.

*Competing interests.* The corresponding author declares that none of the authors has any competing interests.

*Acknowledgements.* We appreciate the assistance of the Hong Kong Observatory (HKO), which provided the meteorological data. The work described in this paper was supported by a grant from the Research Grants Council of the Hong Kong Special Administrative Region, China (Project Nos. C6026-22G and T31-603/21-N).



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
