# Peer review of "Coupling the TKE-ACM2 Planetary Boundary Layer Scheme with the Building Effect Parameterization Model"

_Geoscientific Model Development, 2024_

## Author Comment (AC1)

Response to Reviewer#1

The responses to the reviewers' comments are highlighted in blue, and the revised text is *italicized*.

Interactive reviewer comment on the manuscript *Coupling the TKE-ACM2 Planetary Boundary Layer Scheme with the Building Effect Parameterization Model*

**GMD-2024-205**

By Zhang et al.

**General considerations**

In this contribution the authors present the coupling approach of an urban 'building effect parameterization' (i.e., the surface exchange parameterization) to a recently proposed (improved) boundary layer parameterization scheme for atmospheric RANS type models. The new coupled scheme is compared to a version with a bulk surface exchange treatment, and to the results from another PBL parameterization (one of the 'standard schemes in the literature), once with the bulk surface exchange and once with the 'building effect parameterization'. The approach is evaluated on two case study scenarios with an idealized surface characterization (regular cubes) and an LES (PALM-4U) as a reference. Then, a month-long simulation for the Pearl River Delta (China) with a number of mega-cities (including Hong-Kong) is performed. Data from three lidars (wind profiles), and 31 surface stations (urban and rural) are used for validation.

The study is well designed, and pretty well described (the 'plan' for the paper is good, and much of what needs to be known, can be found somewhere). I cannot say, however, that it is well written (I have added quite a number of 'detailed comments' – mostly related to language or formulations, etc.). A serious language/style update by a native speaker would certainly greatly improve the value of the paper.

Even if I have labelled one of the comments as 'major', I think its resolution is straight forward – so that I can recommend the paper to be published subject to minor modifications.

Major comments

1. Real case simulations & data: the presentation of the data is not complete. The lidars, when introduced should be characterized (urban, rural) and some basic information

on vertical resolution and accuracy should be provided. Also, for the surface stations, information should be provided on the explicit meaning of the different LCZ classes ('compact high rise, LCZ1, etc.), and how many are available for each type (how many urban, how many non-urban), what 'G class' (e.g., Fig. 17, 18, ..) means. Much of this can be found somewhere (I can, for example add the different numbers in each panel in Fig. 17) but the authors could support the reader in providing this information. Furthermore, Figs. 14-16 have 10 urban classes, plus 'water cells' plus 'natural cells', while Figs. 17-19 have 7 urban classes (the remaining three are probably not available) plus 'G stations' and 'rural stations': how do the latter translate to the water cells and natural cells? I suggest to add a sub-section in Section 2 with some of this information.

Dear Reviewer:

Thank you for your thorough review and valuable feedback. We appreciate your positive remarks regarding the design and structure of our study. We acknowledge the need for improvements in language and style, and we are committed to addressing these issues. We have responded to your detailed comments to enhance clarity and coherence. Additionally, we have appointed a native English speaker from AsiaEdit (https://asiaedit.com/) to assist with a comprehensive language and stylistic revision. We are grateful for your constructive input and are pleased that you find the main issues straightforward to resolve.

Firstly, we have provided explanations for the Local Climate Zones (LCZ) codes 1 to 10 and A to G in Table B1. In the revised manuscript, we maintain consistency by avoiding quick switches between terms such as 'LCZ G' and 'water surface', as well as 'natural' and 'rural' by using 'water surface' exclusively to refer to 'LCZ G' and 'rural land' for other non-urban land types.

Furthermore, we have introduced the characterization of wind speed LiDAR units in Section 2.4.1, where essential details, including resolution, accuracy, and operating frequency, are described. Additionally, we clarify that the land cover type for each LiDAR unit is determined by the LCZ classification of the nearest model grid. For instance, the King's Park LiDAR is located within an LCZ 1 grid, and is therefore abbreviated as KP_LCZ1.

Similarly, we have added a brief introduction to the surface stations data obtained from the Global Telecommunication System. The characterization of these surface stations is also informed by the LCZ code of the nearest model grid. Furthermore, we have explicitly listed the number of available stations for each LCZ classification, as detailed in Table B1.

**Table B1.** Local Climate Zones (LCZ) classification scheme.

| LCZ code | Built type | Number of available surface stations | LCZ code | Land cover | Number of available surface stations |
|---|---|---|---|---|---|
| LCZ 1 | Compact high-rise | 2 | LCZ A | Dense trees | 4 |
| LCZ 2 | Compact mid-rise | 1 | LCZ B | Scattered trees | 0 |
| LCZ 3 | Compact low-rise | 0 | LCZ C | Bush and scrub | 3 |
| LCZ 4 | Open high-rise | 3 | LCZ D | Low plants | 0 |
| LCZ 5 | Open mid-rise | 1 | LCZ E | Bare rock or paved | 0 |
| LCZ 6 | Open low-rise | 2 | LCZ F | Bare soil or sand | 1 |
| LCZ 7 | Lightweight low-rise | 0 | LCZ G | Water surface | 10 |
| LCZ 8 | Large low-rise | 3 | | | |
| LCZ 9 | Sparsely built | 0 | | | |
| LCZ 10 | Heavy industry | 1 | | | |
| **Subtotal of urban stations** | | 13 | **Subtotal of non-urban stations** | | 18 |

**Minor comments**

l.48 '...they work with few...': maybe better 'they have only been coupled to a few ... (I think they would also work with all the other schemes – btu it has not been done)

Thank you for highlighting this point. Line 46 has been revised to:

*'However, multi-layer BEP/BEP+BEM models are adopted less widely than the Bulk scheme or SLUCM because they have only been tentatively coupled to a few planetary boundary layer (PBL) schemes [e.g., Boulac (Bougeault and Lacarrere, 1989), MYJ (Janjic, 1994), and YSU (Hong et al., 2006) added recently by Hendricks et al. (2020)]. This is primarily due to ´ the challenges associated with incorporating the transformation of mean kinetic energy into TKE within a first-order closure PBL scheme, such as the YSU scheme. '*

l.60 'have shown that the TKE-ACM2....'

Thank you for the careful evaluation of the manuscript. We have revised the wording in line 69 accordingly as:

*'They showed that the TKE-ACM2 outperformed two other operational PBL schemes, Boulac (Bougeault and Lacarrere, 1989) and ACM2 (Pleim, 2007b), in simulating the vertical profiles of wind speeds.'*

l.62 'at *the* urban station....': this suggests that the reader knows which urban station is meant. Please rephrase.

We have rephrased the sentence in line 71 as:

*'However, overestimated wind speeds persisted throughout the entire surface layer at stations classified as urban type, probably due to the discrepancy resulting from the Bulk parameterization of surface layer fluxes.'*

l.72 '...from *the* high-resolution lidar': same as before (this suggests that the lidar had been introduced before). Reformulate to '....from a high-resolution lidar'.

'...from the high-resolution LiDAR' has been reformulated to:

'... *from a network of high-resolution wind speed LiDAR units*'

l.87 Energy conserving

Revised accordingly.

l.84 '...K is the eddy viscosity....'. Do I have to assume that K is equal for all 'zeta' (l. 91). If not (what would be better supported by the literature) , K should also get an index zeta.

We understand that this expression may cause confusion to readers who might assume K is the same for scalar and momentum. We have revised the expression in the equation for K such that $K_\zeta = K_h$ if $\zeta \in [\theta, q]$ and $K_\zeta = K_m$ if $\zeta \in [u, v]$. Eqns.2 and 3 are revised accordingly.

$$\overline{w'\zeta'_I} = -K_{\zeta,I}\frac{S_I(\zeta^n_{i+1}-\zeta^n_d)}{V_i\Delta z_I} + \text{Mu}(h-z_I)(\zeta^n_1 - \zeta^n_i) \quad \text{Eqn.2}$$

$$\frac{\zeta^{n+1}_i - \zeta^n_i}{\Delta t} = \underbrace{f_\text{com}\,\text{Mu}\zeta^n_1}_{\text{upward cowvective traspport}} \underbrace{- f_\text{conv}\,\text{Md}_i\,C^n_i + f_\text{conv}\,\text{Md}_{i+1}\zeta^n_{i+1}\frac{\Delta z_{i+1}}{\Delta z_i}}_{\text{donwand tranport}}$$

$$+ \underbrace{(1-f_\text{coev})\frac{1}{V_i\Delta z_i}\left[S_I\frac{K_{\zeta,I}(\zeta^n_{i+1}-\zeta^n_i)}{\Delta z_I} - S_{I-1}\frac{K_{\zeta,I-1}(\zeta^n_i-\zeta^n_{i-1})}{\Delta z_{I-1}}\right]}_{\text{local tramport}} + \underbrace{\frac{F_i}{\Delta z_i}}_{\text{env. forcing}} \quad \text{Eqn.3}$$

$K_h$ and $K_m$ is likely to differ in their magnitudes, especially when there are convective thermals. The evidence is given in very detail in Li (2019) which is properly cited. $K_m$ is related to $K_h$ by $K_h = K_m/\text{PR}_t$ where $\text{PR}_t$ is the turbulent Prandtl number. The parameterization of Prandtl number adopted in this study is consistent with that in TKE-ACM2 (Zhang et al., 2024) which is the Businger's relationship (Businger et al., 1971).

The text has been revised in line 130 to:

*'The eddy diffusivity is equal in magnitude for scalars (Kh) and TKE (Ke) and is related to eddy viscosity (Km) through the turbulent Prandtl number (Prt), which is a key parameter pertinent to heat transfer (Li, 2019):*

$$K_e = K_h = K_m/Pr_t$$

*where $K_m = C_K l_k e^{1/2}$, $C_K$ is a $\mathcal{O}(1)$ empirical constant, the parameterization of $Pr_t$ is consistent with that in Zhang et al. (2024) which follows Businger et al. (1971), and ...'*

l.115    'C_eps is an empirical constant and l_eps corresponds to....'

Line 122 is revised as:

*'... $\epsilon = \rho C_\epsilon e^{3/2}/l_\epsilon$ represents the TKE dissipation rate where $C_\epsilon$ = 1/1.4 is an empirical constant and l$\epsilon$ corresponds to the characteristic length of energy-containing eddies.'*

l.162    uniformly distributed in the vertical: this may be a good idea in a CBL but how about the near surface?

When a multi-layer UCM is turned on in WRF, e.g., BEP, the interpolation of prognostic variables ($\zeta$) below the first half eta level such as $U_{10}$, $T_2$ still follows M-O similarity theory (MOST) as if in Bulk simulations, despite it is doubtful that MOST is better justified in the roughness sublayer than in the urban canopy layer. The deviation of MOST and explicitly resolving the height below $1\Delta z = 12.5$ m is not discussed in this study. However, according to Shin & Dudhia (2016), a vertical resolution $\leq 20$ m is deemed as a high resolution in a mesoscale model configuration. The comparison between LES and WRF+BEP is made by linearly interpolating the finer LES grids to the coarser WRF+BEP grids.

In the real case simulations of the present study and in fact in many other studies, e.g., Bhautmage et al. (2022), Shen et al. (2019), the first half eta level corresponds to roughly 9-13 m, which maintains a reasonable balance between the computational cost and accuracy in the urban canopy layer.

l.164    '...one corresponding to a moderately...'

Revised accordingly.

Fig. 2, caption: the different types of lidars should be referenced (UTSS, HT, KP), and briefly explained (possibly in the text) what their strengths weaknesses are.

The subsection (Section 2.4.1) is revised to describe details of instrumentation of LiDAR units deployed at different sites in Hong Kong including USTSS_LCZ5, HT_rural, and KP_LCZ1.

l.215    '...it is found that quasi-....'

Revised accordingly.

l.215    '....when LES...': how is the time for having reached quasi-equilibrium diagnosed?

Shin & Dudhia (2016) found that the TKE increases in time until 1 hr and stops growing after that, supporting that LES has reached a quasi-equilibrium state. In their study, the time scale 1hr corresponds to approximately $6\tau$ where $\tau = PBLH/w_*$ is the large eddy turnover time. Similarly, a factor of 5 is found in other literature, e.g., Ayotte et al. (1996), Pleim (2007). In this study, the time series of TKE exhibits a similar trend to aforementioned studies, where TKE reaches a maximum, followed by a slight descend, and stops growing after that. The maximum vertical velocity shows a similar trend. Ultimately, we found that a time scale corresponding to approximately $10.2\tau$ shown by the vertical dotted lines in the time series could be a critical value for determining LES has reached the quasi-equilibrium. After $10.2\tau$, although the time series exhibits fluctuations in the magnitudes, the instantaneous value does not show considerable deviation from the mean. There is no particular algorithm for diagnosing this time scale. Instead, due to the inability to store LES output at each time step, we found that domain averaged profiles after spinning up approximately $10.2\tau$=6,300 s and 4,200 s in two cases would be appropriate to drive the WRF simulations.

l.223    usually called 'turbulent fluxes'. However, it would be better to delete 'outputted' - these are just the 'turbulent fluxes from PALM'.

Thanks for your careful proofreading. We have corrected 'turbulence fluxes' to '*turbulent fluxes*'. Also, we have deleted the word 'outputted' from the sentence.

l.224    very often, what we can see in a figure has been plotted... (so, the verb 'to plot' is somewhat obsolete in this context). May be '....schemes are contrasted in ...'.

Thanks for the suggestion. We have revised the sentence in line 238 to:

'*The horizontally averaged u and θ profiles during the last 6τ are displayed in Fig.4 and the turbulent fluxes from PALM and computed from WRF PBL schemes are contrasted in Fig.5.*'

Fig. 6, caption: please add for which case this RMSE is determined and what 'the truth' is (assumed to be).

We have revised the caption of Fig.6 to:

'*a-d): RMSE for $\overline{w'\theta'}, \overline{w'u'}, \theta$ and $u$ calculated below the PBL height for Case 10WC by taking the LES results as the ground truth, respectively; e-h): same as a-d) but for Case 24SC*'

Fig. 6: I would find it 'more convincing' if the black dots would be displayed as a 'dotted line' (and not as a dot at each level) – then it would not appear as a black line in the lower parts of the panels......

Thanks for the suggestion. I assume you mean Fig.4 (u and theta comparison) and Fig.5 (turbulent fluxes comparison) but not Fig.6 as Fig.6 is a bar plot and has no dot plots. We have now revised Fig.4 and Fig.5 such that the black dots representing LES results are connected through solid lines.

1. 231 'a smaller warm bias' would possibly sound better

Revised accordingly.

l..234 'becomes stable....': this is indeed a feature of the CBL. Some authors have even defined a 'neutral level', i.e. the height where slightly unstable transits into slightly stable (formally, there might even be such a height in the LES)

Thanks for the comments.

l.241 '....within *[the]* UCL and near *[the ]*PBL height where the relatively constant w'θ' in the middle UCL is not exhibited [reproduced?] in either BEP simulation.'. Here, I think this is a little 'underselling' the BEP simulations. They at least to some degree reproduce a strong deviation in the profile at canopy height (the two others cannot reproduce this), the relax in the middle of the CBL (and yes, the vertical gradient is too small).....

Thanks for pointing this out. We have rephrased line 265 to:

'Greater discrepancies in the magnitude of $\overline{w'\theta'}$ were observed in TKE-ACM2+BEP within the UCL and near the PBL, height where the relatively constant $\overline{w'\theta'}$ in the mid-UCL was not reproduced in either BEP simulation; however, the drastic reduction in $\overline{w'\theta'}$ at roof level was well captured, indicating that the physical interaction with buildings was reasonably considered.'

l. 249 'This has shown the wind shear at the roof level is underestimated...': I am not sure what the authors want to say with this. Maybe this is just a matter of wording? – 'thus it appears that the BEP parameterization results in an underestimation of wind shear at roof level, when compared to the LES'.

Indeed, we aimed to convey a plain fact. We have rephrased line 254 to:

*'It should be highlighted in Fig.4b that from the ground level to the top of the UCL, both BEP simulations overestimated the wind speed in contrast with an underestimation in the mixed*

*layer. It thus appears that the BEP parameterization resulted in an underestimation of wind shear at roof level when compared with the LES.'*

l. 250 'it is discovered…': first of all I suggest to start a new paragraph. Second, momentum flux decreases (increases in magnitude…) with height. Third, 'at some height' (as it appears in the LES) seems to be some 2-4 canopy heights (in b) and d), respectively). Fourth, this cannot be called 'discovered' here – this was even one of the reasons for the development of the BEP scheme (i.e., that it had been discovered earlier, that momentum flux was not constant with height in urban canopies).

Thanks for the detailed comments. First, in the revised manuscript, we have started a new paragraph. Second, we corrected the momentum flux decreases from the ground (not increase). Third, we have substituted 'at some height' to 'at approximately 2 to 4 times the canopy height'. Fourth, we rephrased the sentence to 'It is observed that …'. Consequently, the whole sentence in line 271 is revised to:

*'The momentum flux decreases from zero at the ground level to a maximum value at approximately 2 to 4 times the canopy height followed by a descending trend in BEP simulations, in contrast to the monotonically descending trend in simulations when the Bulk method was adopted as shown in Fig.5a.'*

l.260 'similar behavior of the two schemes is found…'

Revised accordingly.

l.275 I think the dashed line is blue in Fig. 5d

Revised accordingly. This typo was due to that we changed the color scheme for all plots suggested by the journal editor to allow readers with color vision deficiencies to correctly interpret the results.

l.279 top of the RSL, rather

Revised accordingly.

Fig 7 I suggest to repeat the definition of delta_U (i.e., BEP-Bulk) in the caption. Same in Fig. 8 for theta

Agreed. As suggested by Reviewer#2 we improved Figs.7, 8, 14, 15, and 16 by contrasting TKE-ACM2 minus Boulac, both with and without BEP. In the caption, we have repeated the definition of $\Delta U(TKE\text{-}ACM2 - Boulac)$ in the captions.

l.300 beginning a new sentence: Figure 8.....

Revised accordingly.

Fig. 9, caption: delete 'plots the'; 'at USTSS, HT and KP': are these locations? I recall too have seen different symbols in Fig. 2 – and thought this to be different types of instruments. I suggest to add an ultra-short sub-section in Section 2, describing the instrument type, vertical resolution and some accuracy statements from the manufacturer.

The words 'plots the' have been deleted.

'USTSS, HT, and KP' are indeed LiDAR units located at different locations.

Section 2.4.1 is revised to contain the abovementioned information:

*'2.4.1 Landuse data and wind LiDAR observation network*

*This study adopted the 17-class LCZ classification scheme (Demuzere et al., 2022) to more accurately capture the highly variable urban morphology within the domain of interest. The distribution of LCZ 1 to 10 (urban) grids and LCZ A to G (non-urban) grids is depicted in Fig.2c. Each class is defined in Table B1.*

*A wind speed Doppler LiDAR network (see Fig.2d) has been operational in Hong Kong since March 2020, continuously monitoring wind conditions and playing a crucial role in validating regional downscaling results. The network comprises three WindCube 100S LiDAR units manufactured by Vaisala. Each unit measures the vertical profile of the wind speed at an elevation angle of 90°. The units measure 25-m intervals starting from 50m above ground level, with an accuracy of <0.5m/s for wind speed and 2° for wind direction. Although each LiDAR outputs data at a frequency of 1Hz, measurements are averaged hourly and archived due to storage limitations. We represent the land cover type of each LiDAR unit using the LCZ classification associated with the nearest model grid following Ribeiro et al. (2021).*

*The LiDAR unit at the Hong Kong University of Science and Technology Supersite (USTSS_LCZ5) is located on the east coast of Kowloon Island, where the nearest model grid center falls within classified as LCZ 5 (open mid-rise). The second LiDAR, installed on the southeastern peninsula of Hong Kong Island (Hok Tsui), is surrounded by natural vegetation and referred to as HT_rural. Lastly, the LiDAR at King's Park in downtown Kowloon, where the average building height is 60m (Kwok et al., 2020), is located within an LCZ 1 model grid (compact high-rise), and designated as KP_LCZ1.*

*In addition to profiler-type observations, we also used measurements of surface meteorological variables, including the 10- m wind speed (U10), 2-m temperature (T2), and 2-m relative humidity (RH2), retrieved from the Global Telecommunication System. The coordinates and LCZ classifications of these surface stations are provided in the supplementary material of Zhang (2024). The surface station dataset comprises a total of 13 urban stations characterized by LCZ classes 1 to 10, along with 10 stations situated on water*

*surfaces, and 8 rural stations on land. The distribution of surface stations across specific LCZ classes is provided in Table B1.'*

l.306    ...the rural lidar station HT (first, I learn now that the different symbols are different sites (see previous comment), but also I learn that at least one of the lidars is 'rural'. Why not giving them an extension in the acronym?

We have renamed HT to HT_rural, USTSS to USTSS_LCZ5, and KP to KP_LCZ1 in the texts as well as in the figures.

l.306    at the LCZ 5 USTSS lidar location: wouldn't it be perfect to add this LCZ information to the section suggested in 'comment to Fig. 9'?

We have added the description of each LiDAR and its surrounding roughness in **Section 2.4.1**.

l.308    'has been reduced': the authors probably mean 'is smaller in the BEP schemes....'

Revised accordingly.

l.309    I don't think there is a Fig. A51... Can the authors adjust?

We have made necessary corrections with numbering when referencing to the supplementary materials.

l.312    starting at an altitude of 50 m agl?

The Vaisala wind LiDAR series WINDCUBE 100S has a blind spot from the ground level to 50m AGL. The first measurement starts from 50m AGL then every 25m.

Fig.10/11/12, captions: are these instantaneous values at the given times or 1-hour averages (in both, the observations and the simulation? Also, the caption may remind the reader that the panels start at 8 pm (why is this so?)

The LiDAR observations are 1-hour averaged values. The Vaisala wind LiDAR series WINDCUBE 100s is tuned to measure wind speeds at a frequency of 1 HZ. The size of raw data for a single LiDAR unit operating for 24 hours is on the order of 1 TB storage, thus the data is processed by taking the hourly average and save 24 times a day. The simulation is the instantaneous value defined at the integer hour in the WRF namelist.

The sequence of subplots starts from 8pm because the model integration starts from 1200 UTC+0 18th July in 2022 to 1200 UTC+0 18th August in 2022, which translates to 2000 local time in Hong Kong (UTC+8). Therefore, the default time stamp after aggregating the results

starts from 20hr UTC+8, then 21, 22, 23, 24, 1, 2, 3, …,18, 19. We have added the following sentence in the captions of Fig.9 and 10. for clarity.

'*The integration is from 2000 UTC+8 on $18^{th}$ July in 2022 to 2000 UTC+8 on $18^{th}$ August in 2022.*'

Fig. 13: RMSE and mean bias of WHAT? What is the data base? What are the 'error bars' referring to?

We meant the RMSE of monthly averaged diurnal variation of vertical profiles of wind speeds from WRF simulations by benchmarking against the LiDAR measurements, so is the mean bias.

The error bars mean the $\pm 1\sigma$ variability of RMSE/ mean bias calculated at 24 hours.

We have revised texts in the caption as:

'*Figure 13. RMSE (a) and MB (b) of the monthly averaged diurnal variation of vertical profiles of wind speeds calculated at the three LiDAR stations for four simulations obtained by taking LiDAR measurements as the ground truth. The error bars represent the $\pm 1\sigma$ variability of the RMSE/Mean bias of a diurnal cycle.*'

l.330     convective thermals

Revised accordingly.

l.331     the smallest RMSE and the smallest negative bias….

Revised accordingly.

l.332     Boulac+BEP, which increased the deviations with respect to the Boulac+bulk simulations.

Revised accordingly.

l.334     I cannot locate Section 44.1. please adjust.

We have removed the duplicated number and revised it as Section 4.1.

l.337     this is not predictability, rather 'accuracy'

We have revised the word 'predictability' as accuracy.

l.342     who is collaborating here with whom?

This is a typo where we meant 'corroborating'. We have revised the sentence in line 360 as:

*'Both schemes coupled with BEP exhibited considerably decelerated wind speeds below ~ 400m, corroborating the trend observed for all LCZ 1 girds shown in Fig.C2a.'*

l.359    as small as…

Revised accordingly.

l.359    '…is more likely to be found at around 06LT in TKE-ACM2….': I don't think this can be said like that. Do the authors want to say that 'delta_U10 starts to be larger (in absolute terms) starting from about 06 LT'?

We have overhauled Section 4.3 according to Reviewer#2's suggestions, where we discarded the comparison between BEP and Bulk that is well-known and clear, rather we now present comparison between TKE-ACM2 and Boulac in BEP/Bulk simulations.

l.366    slightly altered?

Revised as 'gently altered' to 'slightly altered'.

l.369    the 'supplementary Zhang (2024) is not a proper citation (in the supplementary material to Zhang….)

We have revised line 387 as:

*'Time series data for each station are provided in the supplementary material of Zhang (2024)'*

l.377    LC1…stations are …lower than the observed values': this is, first of all, not a correct sentence (the simulated wind speed at these stations is smaller than…). Second this is a very important observation, which suggests that the authors should (maybe in the appendix) produce a table where the LCZ codes are described in words (having read the sentence, I, for example would wonder what LCZ2 is (it is also having much lower wind speeds than observed….). I suggest to add this finding explicitly to the conclusions (in the present form it states that LCZ1,4, 10 etc. are underestimating – but it is more relevant to state that high-rise and heavy industry types are underestimating.

Thanks for your suggestion in highlighting our key findings. We have revised the text as:

*'However, the simulated wind speeds simulated using BEP at LCZ 1, 2, 4, and 10 stations were lower than observed values, particularly during the day.'*

Second, we have added in the appendix outlining the code for each LCZ class along with brief description.

Third, we have highlighted explicitly that BEP simulations lead to underestimation in high-rise and industry type grids in the conclusion in line 446 to line 449:

*'BEP did not necessarily improve the prediction of U10 at all types of urban stations as it could lead to largely underestimated U10 relative to the two schemes with Bulk methods. For instance, extremely low wind speeds were observed at LCZ 1, 2, 4, and 10 stations, which were in areas that had mostly compact or high-rise buildings. The enhanced accuracy of U10 simulated by TKE-ACM2+BEP was notable at stations located in areas of relatively low building density, such as LCZ 5, 6, and 8 stations.'*

l.383    'at the hill whose.....': replace by 'at a hill with a spatial scale of 50 m'.

We have revised the wording in line 399 as:

*'For instance, the surface station co-located with the KP_LCZ1 LiDAR, also classified as an LCZ 1 station, was situated on a hill with a spatial scale of 50m.'*

l.388    'Coinciding with Fig. 15'? Maybe: 'As can be seen in Fig. 15, T2....'?

We have revised line 381 to '*Figure 15 shows that the temperature difference ΔT2(TKE-ACM2 − Boulac) followed a diurnal pattern, with TKE-ACM2 consistently simulating lower T2 at 12LT relative to Boulac which...*'.

l.391    'their predictability': it is accuracy and not predictability

The word 'predictability' is replaced by '*accuracy*'.

Figs17-19: what are 'G' stations?

We have added the code of LCZ class in the appendix as mentioned previously. In addition, we have replaced 'LCZ G' by explicitly referring to 'water surface'. We avoided frequent switching of wordings of 'LCZ G' and 'water surface' in the revised manuscript, instead, we used 'water surface' exclusively in the texts.

l.400    again, it is not the predictability that is improved, but the prediction (i.e., its accuracy). Predictability is a property of the atmosphere (which is assessed using ensemble prediction approaches)

Revised accordingly. The word 'predictability' has been revised in other places in the manuscript.

l.401    should read: ....BEP produces larger RH2 when ....

Line 420 is revised as:

*'Figure C5 shows that BEP produced an increasingly large RH2 when coupled with TKEACM2 rather than with Boulac, resulting in a more profound improvement in TKE-ACM2+BEP.'*

l.409    building-resolving

Revised accordingly.

Figure 20, caption: Please add the information (in the caption) where the number of sites contributing to a LCZ type can be found.

We have revised the caption of Figure 20 as:

*'RMSE for aggregated station types, with a), b), and c) representing U10, T2, and RH2, respectively. The number of stations contributing to an LCZ type is given in the sub-titles in Fig.17, Fig.18, or Fig.19.'*

l.419    BEP suggests that the buildings act..

Revised accordingly.

l.421    ...observations are used to...

Revised accordingly.

l.425    ...LIDAR station, compared to..

The comma has been before 'compared to'.

l.430    no predictability

The word 'predictability' has been replaced with 'accuracy'.

References

Ayotte, K., Sullivan, P., Andren, A., Doney, S., Holtslag, B., Large, W., McWilliams, J., Moeng,
        C.-H., Otte, M., Tribbia, J., & Wyngaard, J. (1996). An Evaluation of Neutral and
        Convective Planetary Boundary-Layer Parameterizations Relative to Large Eddy
        Simulations. *Boundary-Layer Meteorology*, *79*, 131–175.
        https://doi.org/10.1007/BF00120078

Bhautmage, U. P., Fung, J. C. H., Pleim, J., & Wong, M. M. F. (2022). Development and
        Evaluation of a New Urban Parameterization in the Weather Research and

Forecasting (WRF) Model. *Journal of Geophysical Research: Atmospheres*, *127*(16).

https://doi.org/10.1029/2021JD036338

Businger, J. A., Wyngaard, J. C., Izumi, Y., & Bradley, E. F. (1971). Flux-Profile Relationships

in the Atmospheric Surface Layer. *Journal of Atmospheric Sciences*, *28*(2), 181–189.

https://doi.org/10.1175/1520-0469(1971)028<0181:FPRITA>2.0.CO;2

Li, D. (2019). Turbulent Prandtl number in the atmospheric boundary layer—Where are we

now? *Atmospheric Research*, *216*, 86–105.

https://doi.org/10.1016/j.atmosres.2018.09.015

Shen, C., Chen, X., Dai, W., Li, X., Wu, J., Fan, Q., Wang, X., Zhu, L., Chan, P., Hang, J., Fan,

S., & Li, W. (2019). Impacts of High-Resolution Urban Canopy Parameters within the

WRF Model on Dynamical and Thermal Fields over Guangzhou, China. *Journal of

Applied Meteorology and Climatology*, *58*(5), 1155–1176.

https://doi.org/10.1175/JAMC-D-18-0114.1

Shin, H. H., & Dudhia, J. (2016). Evaluation of PBL Parameterizations in WRF at

Subkilometer Grid Spacings: Turbulence Statistics in the Dry Convective Boundary

Layer. *Monthly Weather Review*, *144*(3), 1161–1177. https://doi.org/10.1175/MWR-

D-15-0208.1

---

## Author Comment (AC2)

Response to Reviewer#2

The responses to the reviewers' comments are highlighted in blue, and the revised text is *italicized*.

**Zhang et al., 2025: Coupling the TKE-ACM2 Planetary Boundary Layer Scheme with the Building Effect Parameterization Model**

The authors present in the manuscript a development and performance of coupling of TKE-ACM2 PBL scheme with BEP urban model in WRF mesoscale model. Although it describes important and interesting topic of improving of WRF model performance, and also the design of the study seems reasonably, the manuscript is not well written. Sometimes it is hardly readable, confused, some parts are too long but other information are missing. The manuscript have to be substantially improved (or re-submitted) before publishing in GMD.

**Specific major comments:**

1/ The text of the manuscript is not well transparent, some results parts are too long, model formulation could be also shorter or moved into the appendix. Some short sections (e.g. 2.4) could be removed and the number of figures reduced. Further, the manuscript is hardly readable due to often quick switching between ideas and also missing links to figures. It seems that it was not preciously revised by authors before submission.

Dear Reviewer,

Thank you for your thoughtful and constructive feedback on our manuscript titled "*Coupling the TKE-ACM2 Planetary Boundary Layer Scheme with the Building Effect Parameterization Model*." We sincerely appreciate the time and effort you invested in reviewing our work. Your insights have been invaluable in guiding our revisions, and we have made significant changes to enhance the clarity and readability of the manuscript.

**Model Formulation**: In response to your suggestion, we have condensed the model formulation section by relocating some detailed derivations to the appendix. The numerical solutions to the prognostic equation (Eqn.3) are now removed from **Section 2.1 Numerical method to couple TKE-ACM2 and BEP** and instead detailed in Appendix A.

**Removal of non-essential results discussion**: Some non-essential parts of the text are removed to help focus on the key findings, e.g., the comparison between BEP and Bulk is removed as it is well investigated.

**Removal of Short Sections**: We have much shortened the introduction of the Local Climate Zones (LCZ) in the original **Section 2.4** without missing the essential information and reducing the reproducibility of the results.

**Readability and Flow**: To address the readability issues, we carefully revised the manuscript to ensure smoother transitions between ideas. For instance, we have revised in line 243 to line 305 where any confusion between the descriptions of Figures 4 and 5 is eliminated. The descriptions have been reorganized to clearly differentiate the two figures, ensuring that each is described in its own context without overlap. Meanwhile, we have included active references in Latex to relevant figures within the text when discussing the results. In addition, we have provided a Table B1 in Appendix B to demonstrate the number of available surface stations in each LCZ classification. We have also renamed the three LiDAR sites from USTSS, HT, and KP to USTSS_LCZ5, HT_rural, and KP_LCZ1, respectively according to Reviewer#1's suggestion to remind readers about the characterization of each LiDAR site and thus improve the readability.

**Careful Revision**: We apologize for any oversight in the initial submission and have conducted a thorough review to ensure the clarity and quality of the manuscript.

2/ Language level is not sufficient, proofreading by English native speaker would be appropriate.

Thank you for your constructive feedback regarding the language quality of the manuscript. We understand the importance of clear and effective communication in presenting our findings. In response to your comments, we have appointed a native English speaker from AsiaEdit (https://asiaedit.com/) to proofread and revise the manuscript. The track changes file shows the corrections to instances where language revision is needed.

3/ Description of model setting is insufficient, BEP parametrization setting of urban canopy parameters in specific LCZ is missing. Author does not consider possible inaccuracy in the setting of such parameters with impact to model performances in specific LCZ.

The BEP parameterization depends on an essential input known as the look-up table for urban morphology parameters (UCP) and thermal and radiative properties (URBPARM_LCZ.TBL when using LCZ). In this study, the look-up table remains as specified in the WRF 4.3.3 GitHub repository. Specifically, thermal properties such as emissivity, albedo, and thermal conductivity retain their default values. Similarly, the distribution of building heights for each LCZ adheres to the default generic values, which are detailed in Table B1 for clarity. These prescribed parameters are consistent with the values recommended by Stewart & Oke (2012).

The major limitation of applying the look-up table method for UCP is that the heterogeneity of UCP for a certain LCZ urban class is not considered, causing it less accurate compared to a gridded UCP approach (Sun et al., 2021). As reported by Shen et al. (2019), one of the crucial UCP, urban fraction, has paramount importance in simulating the horizontal wind

speeds. However, the variability of urban fraction or building height distribution for a certain LCZ urban class is not taken into account in the present study.

In additoin, the process of re-gridding the LCZ global map from a 100-m resolution to a 1-km model cell raises concerns about the accuracy of the represented LCZ types (Ribeiro et al., 2021a; Sun et al., 2021). This challenge is further compounded by discrepancies between the land use at local observation stations and the land use depicted by the 1-kilometer model grid. Consequently, the UCP assigned to a specific LCZ type may lack adequate representativeness, especially when a model cell encompasses a variety of LCZ constituents, resulting in an absence of sub-grid variability.

We have added the abovementioned potential uncertainties in line 401.

4/ Description of LIDAR and station data is incomplete. Some special section about observation data is usual in papers, with information about measuring sites, variables, locations and other important characteristics in view of comparison with model data.

We have included an introduction to the LiDAR instrument in the revised **Section 2.4.1**, detailing its resolution, accuracy in measuring wind speed and direction, and operating frequency. Additionally, we have described the characteristics of the three sites where the LiDAR units are installed. Finally, we clarified that the classification of the measurement sites is based on the LCZ landuse associated with the nearest model grid following Ribeiro et al. (2021). Likewise, we have introduced that the surface station data is retrieved from Global Telecommunication System, where the method of classification of each station follows that of the LiDAR unit.

5/ Arrangement of Fig. 7, 8, 14, 15 and 16 shows rather impact of BEP urban scheme compared to Bulk, what is clear and well known fact, but not the impact of TKE-ACM2 PBL scheme compared to Boulac, which is the topic of the paper. Differences between simulations with/without TKE-ACM2 scheme should be rather displayed and also impact of TKE-ACM2 scheme more discussed.

We appreciate your guidance in helping us improve the alignment of our figures with the paper's core topic. We have made the following revisions according to your feedback: Figures 7, 8, 14, 15, and 16 have been updated to display the differences between the TKE-ACM2 and Boulac PBL schemes, both with and without the inclusion of the BEP urban scheme. Meanwhile, discussions in Section 4.1 and Section 4.3 are overhauled. This comparative analysis effectively highlights the impact of the TKE-ACM2 scheme, thereby reinforcing the focus and objectives of this study.

6/ High number of mistakes, typos, wrong use of dashes and connectors (see below). I would recommend to authors to use latex with active references for all figures, sections and tables, to enable better orientation in the text (showing of references by click on) and to prevent mistakes in numbering of figures, sections and tables.

Thanks for the careful evaluation. We have addressed all the identified mistakes in numbering and corrected the use of dashes and connectors. We also improved the phrasing and wording with the assistance from the native English proofreader. Additionally, we have ensured clickable references for all figures, sections, and tables are properly compiled in Latex. Detailed corrections can be found in the response below or in the track changes file.

**Other comments and technical corrections:**

L 12 – comparison to Bulk method, similarly L 27, that's not clear if Bulk is meant as some simple PBL scheme or simple urban scheme

Thanks for your comment. We have clarified the meaning of Bulk as 'without any urban scheme'. The sentence in line 10 has been revised to:

'*High-resolution wind speed LiDAR observations suggest that TKE-ACM2+BEP reduces overestimation in the lower part of the boundary layer compared with the Bulk method, which lacks an urban scheme, at a LiDAR site located in a densely built environment.*'

L 19 – brace near brace doesn't look well (L 32)

We have revised line 19 and line 33 as follows:

Line 19: … 'and the overlying roughness sub-layer, or RSL (Rotach, 1999).'

Line 32: 'The single-layer urban canopy model (SLUCM) pioneered by Kusaka et al. (2001); Kusaka and Kimura (2004) is …'

L 23 – 10-50

We have revised '10-50' to '10 to 50'.

L 35 – mathematical formula as Fi is superfluous in introduction (similarly L 41)

Agreed. We have removed the mathematical representation of multi-layer fluxes Fi in the introduction.

L 44 – what is urban heat island circulation? UHI or circulation in urban areas.

We have rephrased 'urban heat island circulation' to urban heat island effect' according to the cited work, i.e., Wang et al. (2017), stating that the urban heat island effect is well captured using BEP/BEM in Hong Kong.

L 48 – braces in braces doesn't look well (L 63 similarly, L 154)

We have revised line 69 as follows:

'*They showed that the TKE-ACM2 outperformed two other operational PBL schemes, Boulac (Bougeault and Lacarrere, 1989) and ACM2 (Pleim, 2007b), in simulating the vertical profiles of wind speeds.*'

We have revised line 154 as follows:

'*The prescribed height of building arrays is justified by that it is commonly seen in Hong Kong according to Kwok et al. (2020).*'

L 49 – word order … added recently by H…

Revised accordingly.

L 51 – motivation better explained

We have added a few sentences from line 48 to better emphasize our motivation in coupling a 1.5-order non-local closure PBL model with the BEP model:

'*However, multi-layer BEP/BEP+BEM models are adopted less widely than the Bulk scheme or SLUCM because they have only been tentatively coupled to a few planetary boundary layer (PBL) schemes [e.g., Boulac (Bougeault and Lacarrere, 1989), MYJ (Janjic, 1994), and YSU (Hong et al., 2006) added recently by Hendricks et al. (2020)]. This is primarily due to ´ the challenges associated with incorporating the transformation of mean kinetic energy into TKE within a first-order closure PBL scheme, such as the YSU scheme. As a result, the eddy diffusivity can only be adjusted in response to surface fluxes, limiting its ability to account for the generation and dissipation of TKE through other boundary layer processes, such as the generation of TKE by wind shear and buoyancy. Additionally, the other two PBL schemes (MYJ and Boulac) model the vertical mixing of momentum between two adjacent layers, but lack the non-local mixing driven by large-scale eddies under convective conditions. For instance, Coniglio et al. (2013) reported that MYJ produces PBLs that are too shallow and moist PBLs in the evening, and Xie et al. (2012) found that the PBL height diagnosed by Boulac may be too short to be realistic.*'

L 160 – the horizontal resolution of WRF+BEP in idealized case is not clear

The horizontal resolution of WRF+BEP was described in line 157:

'*WRF+BEP runs at ta building-parameterized scale ($\Delta x = \Delta y = 1$ km)*'

L 175 – WRF+BEP other setting is not described

The configuration of idealized WRF+BEP in Section 2.3 (line 149) is rather simplistic because the simulations are prescribed with idealized initial and boundary conditions, where physics such as microphysics and radiation scheme are turned off.

Specifically, the initial condition of wind speed is described in line 161 with the Coriolis parameter being $10^{-4}s^{-1}$ and that of potential temperature has analytical expression following Eqn.10 and Eqn.11. The landuse of all model cells is prescribed as urban type. The parameterization of cumulus and microphysics are turned off in WRF+BEP to keep it consistent with the LES setting. The short/long wave radiation schemes are also turned off because the net heat flux is prescribed with user-specified values. Additionally, the land surface model/surface layer schemes are not used for calculating surface fluxes for the same reason. However, the namelist options for these two physics (sf_surface_physics and sf_sfclay_physics) are still assigned with 8 and 1, respectively, otherwise BEP subroutines cannot be called.

An implication of this idealized configuration is that the thermal properties of buildings and streets specified in the look-up table (URBPARM.TBL) become ineffective because the radiation transfer is essentially prescribed by the idealized heat flux. The key parameters defined in the look-up table are a uniform building height of 40 meters, a street width of 30 meters, and a building width of 20 meters which are reported in the manuscript.

Chap. 2.4 – why is it separated? Is is used in idealized case, or is it belonging rather to real case?

We have reorganized Section 2.4.1 to describe the Local Climate Zones (LCZs) used in real case simulations. In addition, we have introduced the wind LiDAR observation network in this section, where we clarified the approach to classify the landuse type of the LiDAR unit.

L 186 – you talk firstly about 10 LCZ and here about 17 classes

The LCZ classification scheme has in total 17 classes, consisting of 10 urban classes and 7 non-urban classes. We have clarified in line 183 as follows:

'*The distribution of LCZ 1 to 10 (urban) grids and LCZ A to G 180 (non-urban) grids is depicted in Fig.2c. Each class is defined in Table B1.*'

Additionally, we have clarified the definition of 17-class LCZ in Appendix B Table B1 where the 10 urban classes along with the 7 non-urban classes are explicitly defined.

L 198 – the formulation "July 18 20 o'clock" is unclear, need reformulate. Similarly the following sentence.

Line 206 has been revised to:

*'30-day simulations are performed between 1200 UTC+0 on 18th July to 1200 UTC+0 on 18th August of year 2022.'*

L 204 – this sentence is without any notice about moving to WRF setting

We have made the introduction to the physics settings of WRF a separate paragraph following the sentence "We used NCEP GFS analysis data at 6-hourly input intervals to provide the initial and lateral boundary conditions."

The separate paragraph in line 213 reads,

*'Identical physics schemes are chosen in the four simulations: unified Noah scheme (Chen and Dudhia, 2001) for the land-surface model, WSM 3-class simple ice scheme (Hong et al., 2004) for microphysics, RRTMG scheme (Iacono et al., 2008) for longwave/shortwave radiation, and Grell-Freitas ensemble scheme (Gall et al., 2013) for cumulus.'*

L 205 – Bulk scheme is usually not considered as a canopy model (UCM), because there is no canopy

Agreed. We have clarified that the Bulk scheme refers to the configuration where the surface layer fluxes are computed using Noah land-surface model without any UCM.

Lines 215 is revised to:

*'The TKE-ACM2 PBL scheme was coupled with the BEP UCM (referred to as TKE-ACM2+BEP) and evaluated alongside the TKE-ACM2 scheme in isolation (TKE-ACM2+Bulk), where the surface layer fluxes were computed using the Noah land-surface model. The Boulac PBL scheme underwent the same evaluation, being coupled with the BEP UCM (Boulac+BEP) and assessed in isolation with the Noah land surface model (Boulac+Bulk).'*

L 215--220 – acronyms are unclear, all sentences should be written better

The mathematical symbols and acronyms are revised and the sentences are re-written in line 233 as:

*'Quasi-equilibrium was achieved in the two LES cases after approximately 10.2 convective turnover times ($\tau$), where $\tau = h/w^*$, and $w^* = \left(\beta \overline{w'\theta'_0} h\right)^{1/3}$ represents the convective velocity scale. The duration of 10.2 large-eddy turnover times is considered a reasonable indicator of well-developed dynamic fields over the domain with buildings, especially when compared to other studies that have used factors of 5 (Ayotte et al., 1996; Pleim, 2007b; Zhang et al., 2024) and 6 (Shin and Dudhia, 2016) for flat domains.*

*The horizontal averages of the velocity and potential temperature fields are calculated at 10.2τ and serve as initial conditions for driving mesoscale WRF simulations for an additional 20τ . Subsequently, the results from the final 6τ, corresponding to either 3600 seconds or 2400 seconds, are averaged both horizontally and temporally. Table 1 summarizes the key turbulence characteristics of the convective flow and the runtime parameters.'*

Fig. 4 – dotted line is not well visible in plots

Thanks for the comment. We have connected the dots representing the LES results with lines.

Fig. 5 – blue dotted and dashed lines are not in the legend

We have added the legend for the TKE-ACM2+BEP momentum flux which consists of the non-local (dashed) and the local (dotted) components in Fig.5.

L 240–250 and further – links to figures are missing, the text is still switching between description of Fig. 4 and 5

We have added necessary links and active references to figures from line 248 To line 309 to ensure that the references are clear and easily accessible for the reader. Additionally, we have revised the text to eliminate any confusion between the descriptions of Figures 4 and 5. The descriptions have been restructured to clearly distinguish between the two figures, ensuring that each figure is described in its own context without overlap.

L 256 – prorportion → proportion

Revised accordingly.

L 274 – is blue dashed line non-local or local component? (sentence vs. Fig. 5 caption), there is also no red dashed line

We have revised Fig.5 and also the texts so that the blue dashed line represents the non-local component and the blue dotted line denotes the local component.

L 275–276 – the sentence not clear

We have revised the sentence in line 295 as follows:

'Compared with Case 10WC, the larger prescribed $\overline{w'\theta'}_0$ in Case 24SC suggests that TKE-ACM2+BEP achieved a closer match in the magnitude and shape of $\overline{w'u'}$ at and immediately above roof level compared with Boulac+BEP. '

L 283 – fount → found

Revised accordingly.

L 283–285 – the sentence is not consistent to claim in L 244. I think a different order of variables in Fig. 4 and 5 vs. Fig. 6 caused it. I would recommend to change the order of variables in Fig. 6

Thanks for the comment. We have revised the order of variables in Fig.6 such that the order follows that in Fig.4 and 5, which reads $\theta, u, \overline{w'\theta'}, \overline{w'u'}$. The sentence in line 303 draws conclusions for Case 24SC where line 252 describes the results for Case 10WC.

To avoid confusion to readers, we have revised line 303 as:

'*This indicates that the two PBL schemes coupled with BEP performed similarly in simulating momentum profiles below the PBL height in Case 24SC and outperformed the Bulk methods.*'

L 289 – what does mean "other natural landuse" – is it any crop, forest or pasture?

We intended for 'other natural landuse' to refer to the landuse that is non-urban (LCZ 1 to 10) and also not water surface (LCZ G). To enhance clarity, we revised all instances of 'other natural landuse' to 'rural land cover' throughout the texts and figures.

L 299 – besides urban grid-boxes, the BEP model is not used over natural and water grid-boxes in simulation, so it cannot produce any direct difference in U, only as an impact of neighbouring grid-boxes

Agreed. We have rephrased the texts in line 311 as:

'*Both BEP simulations had less pronounced differences in U over water surfaces and rural land cover compared with urban grids, primarily because the BEP model was not directly applied in these non-urban areas. Any observed differences in U in these regions resulted from the neighboring urban grids.*'

L 302 – the sentence is not correct, there are other mechanisms except anthropogenic heat (e.g. shadowing of solar radiation by buildings), which cause lower temperature in BEP simulation in comparison to Bulk

Agreed. We realize that the total heat flux can consist of shortwave/longwave radiation received by the surface, and sensible heat through conduction computed in BEP, ultimately resulting in the lower temperature in our case. We have rephrased the sentence in line 319 as:

'*Finally, complex interactions between the atmosphere and buildings, including radiative transfer (direct and reflected solar radiation and net longwave radiation), and thermal exchange between solid surfaces and the atmosphere, collectively led to the lower temperature in BEP simulations.*'

Fig. 7 – there is no direct comparison of model and observation data

In response to your major comment #5, Figures 7 and 8 now focus on illustrating the impact of TKE-ACM2 on the vertical profiles of potential temperature and wind speed as compared to Boulac, both with and without the BEP. Additionally, the effects on 10-meter wind speed, 2-meter temperature, and 2-meter relative humidity are depicted in Figures 14, 15, and 16, respectively. This shift in focus better aligns with the core topic of our paper, moving away from the well-known comparison between BEP and Bulk methods.

We have also incorporated a comparison with observational data, including wind speed LiDAR measurements and surface station data, as detailed in Sections 4.2 and 4.3. Given the relatively sparse distribution of observation units in relation to the model grid, simulations are evaluated exclusively at grid points that encompass any measurement stations.

L 309 – there is no Fig. A51 in the manuscript

This sentence is deleted to avoid confusion.

L 331 – rather "the lowest RMSE and the lowest negative MB"

Revised accordingly.

L 335 – there is no Section 44.1

We have removed the duplicated number and revised as 'Section 4.1'.

L 369 – in the supplementary Zhang (2024) → rather in the supplementary material of Zhang (2024) … or similarly

We have revised it as '… in the supplement material of Zhang (2024)'.

Fig. 17, 18 and 19 – it is not clear, how the stations are assigned to LCZ. Fig. 2 shows only 10 urban stations, but Fig. 17 etc. computes with 23 stations in urban areas.

The classification of each surface station is determined by the LCZ landuse of the nearest model cell center following Ribeiro et al. (2021). There are in total 13 urban stations (LCZ 1 to 10) and 18 non-urban stations (LCZ A to G) in the finest domain 4 (1 km resolution). The breakdown of all types of stations is listed below:

| LCZ classification (urban) | Number of stations | LCZ classification (non-urban) | Number of stations |
|---|---|---|---|
| 1 Compact high-rise | 2 | A Dense trees | 4 |
| 2 Compact mid-rise | 1 | B Scattered trees | 0 |
| 3 Compact low-rise | 0 | C Bush and scrub | 3 |
| 4 Open high-rise | 3 | D Low plants | 0 |
| 5 Open mid-rise | 1 | E Bare rock or paved | 0 |
| 6 Open low-rise | 2 | F Bare soil or sand | 1 |
| 7 Lightweight low-rise | 0 | G Water surface | 10 |
| 8 Large low-rise | 3 | | |
| 9 Sparsely built | 0 | | |
| 10 Heavy industry | 1 | | |
| **Subtotal** | **13** | **Subtotal** | **18** |

The table above is included in Table B1 for clarity.

**Figure 2d** illustrates the distribution of surface stations, represented by blue and red circles. However, some stations overlap, making it difficult to assess their individual locations. To enhance clarity, the figure below focuses solely on the distribution of surface stations by removing the LiDAR units. This revised visualization clearly shows a total of 13 urban stations (red) and 18 non-urban stations (blue).

[Figure]

The following text has been added to clarify how the LiDAR unit and surface station is classified: '*We represent the land cover type of each LiDAR unit using the LCZ classification associated with the nearest model grid following Ribeiro et al. (2021).*' in line 186.

L 382 – reported in (Ribeiro et al., 2021) – wrong braces

We have revised line 399 as '*...reported by Ribeiro et al. (2021)*'.

L 400 – "influence of BEP is relatively marginal on RH 2 at non-urban stations" this is quite trivial meaning when BEP is not operating in non-urban grid-boxes.

Agreed. We have deleted the non-essential information.

L 419 – "BEP indicates the buildings act as a sink of heat" – I think this is not a correct statement, there is no sink of energy, the reasons for lower temperature under BEP are different.

Thanks for the careful evaluation. We are aware that the lower temperature simulated by BEP is a net effect from incoming and reflected shortwave radiation, received and outgoing longwave radiation, and conduction between the atmosphere and buildings. The phrasing of the sentence in line 437 has been changed to:

' *Likewise, the effects of BEP considering the radiative transfer and sensible heat fluxes between solid surfaces and the atmosphere ultimately led to a lower θ over all urban grids.*'

References

Ribeiro, I., Martilli, A., Falls, M., Zonato, A., & Villalba, G. (2021a). Highly resolved WRF-BEP/BEM simulations over Barcelona urban area with LCZ. *Atmospheric Research*, *248*, 105220. https://doi.org/10.1016/j.atmosres.2020.105220

Ribeiro, I., Martilli, A., Falls, M., Zonato, A., & Villalba, G. (2021b). Highly resolved WRF-BEP/BEM simulations over Barcelona urban area with LCZ. *Atmospheric Research*, *248*, 105220. https://doi.org/10.1016/j.atmosres.2020.105220

Shen, C., Chen, X., Dai, W., Li, X., Wu, J., Fan, Q., Wang, X., Zhu, L., Chan, P., Hang, J., Fan, S., & Li, W. (2019). Impacts of High-Resolution Urban Canopy Parameters within the

WRF Model on Dynamical and Thermal Fields over Guangzhou, China. *Journal of Applied Meteorology and Climatology*, *58*(5), 1155–1176. https://doi.org/10.1175/JAMC-D-18-0114.1

Stewart, I. D., & Oke, T. R. (2012). Local Climate Zones for Urban Temperature Studies. *Bulletin of the American Meteorological Society*, *93*(12), 1879–1900. https://doi.org/10.1175/BAMS-D-11-00019.1

Sun, Y., Zhang, N., Miao, S., Kong, F., Zhang, Y., & Li, N. (2021). Urban Morphological Parameters of the Main Cities in China and Their Application in the WRF Model. *Journal of Advances in Modeling Earth Systems*, *13*(8), e2020MS002382. https://doi.org/10.1029/2020MS002382

Wang, Y., Di Sabatino, S., Martilli, A., Li, Y., Wong, M. S., Gutiérrez, E., & Chan, P. W. (2017). Impact of land surface heterogeneity on urban heat island circulation and sea-land breeze circulation in Hong Kong. *Journal of Geophysical Research: Atmospheres*, *122*(8), 4332–4352. https://doi.org/10.1002/2017JD026702

---

## Author Response (AR2)

Reviewer comment on the manuscript

Coupling the TKE-ACM2 Planetary Boundary Layer Scheme with the Building Effect

Parameterization Model

GMD-2024-205

By Zhang et al.

General considerations

The authors have made great efforts to react to the first reviews' comments. This is particularly

true with respect to language and clarity of the presentation. I nevertheless have a number of

minor comments that I feel should be addressed before the paper can be recommended for

publication.

We truly appreciate your insightful comments and constructive feedback, which have

significantly contributed to improving the quality of our work. In response to your comments

and suggestions, we have made point-to-point modifications shown below and addressed them

in the manuscript correspondingly.

Minor comments

(all lines refer to the 'track changes' manuscript [gmd-2024-205-ATC3.pdf])

1. 65 '....PBLs that are too shallow and moist in the evening...'

The repeated "PBLs" is deleted. The sentence is revised to:

For instance, Coniglio et al. (2013) reported that MYJ produces PBLs that are too shallow and

moist in the evening.

eq. (8) and (9). I overlooked this in the first version. There seems to be a mismatch in the use

of variable name 'z' and 'z\_i'. You have introduced 'z\_i' in eq (2) as the level heights in relation

to the discrete parameterization for the fluxes. Here, a general profile is defined as a function

of the height coordinate 'z'. Therefore, all heights in these equations should read 'z', and not

'z\_i'. (this is particularly important because your case is a CBL, where usually the height of the inversion (CBL height) is denoted 'z i'.

Indeed, the original notation of 'z\_i' could cause confusion with the CBL height. We have removed the subscript 'i' in equations 8 and 9.

1. 230 '...in Table B1': here, it must become clear that this table is in Appendix B.  $\rightarrow$  '....in Tab. B1 (Appendix B)'.

Thank you for the carefulness. We have revised it as follows:

"The distribution of LCZ 1 to 10 (urban) grids and LCZ A to G (non-urban) grids is depicted in Fig.2c. Each class is defined in Table B1 (Appendix B)."

1. 253 'are provided in the supplementary material of Zhang (2024)'. If it is written like this, I assume that this is the supplementary material to https://doi.org/10.1029/2023JD040432, which is the paper on the turbulence parameterization TKE-ACM2. However, it seems that here, the supplementary material to the present paper is meant (i.e., the Zenodo publication of the data etc.). If so, the reference should read: ,....is published on a data hub (Zhang et al 2024)'. Applies also to later occurrences of 'supplementary material'.....

We indeed intended to refer to the supplementary material to the present paper. Thus, we have revised the wording as:

"The coordinates and LCZ classifications of these surface stations are provided in the supplementary material to the present work, which is published on a data hub (Zhang, 2024)."

Any other instances have also been corrected.

1. 256 same as 1.230.

Revised accordingly.

1.320 ....the boundary layer became sightly unstable....

Revised accordingly as follows:

"... and a discrepancy in the Boulac+BEP results relative to the LES results suggests that the

boundary layer became slightly unstable from approximately ~10H."

1.322 '....was well simulated at roof level in BEP and PALM...': but PALM is the reference

(the truth) here – so, PALM only 'well represents the expectations...'. Rephrase to the end that

'BEP quite successfully reproduces the PALM reference'.

It has been revised to:

"Figure 4b suggests that PALM simulated a strong wind shear at the roof level, while such an

inflection point in the wind speed profile was successfully reproduced by BEP, in contrast with

the Bulk simulations,..."

1.327 ...in contrast to...

Revised accordingly.

Fig. 6, caption: please replace 'ground truth' by 'reference'.

Revised accordingly.

1.333 '...leading to conduction': I overlooked this is the first review, but heat conduction is

likely not the reason for this behaviour. I would rather assume this to be a model artifact....(in

the LES). This also concerns the following sentence... (I am not sure whether the PALM model

includes conduction). Pease clarify.

The sensible heat flux (HF) between the solid surfaces (roof, wall, and streets) and the

atmosphere occurs due to their physical contact, which is a means of conduction. However, the

parameterization of HF between the horizontal surface (HF hor) and air is different to that

between the vertical surface (HF vert) and air.

Below the building height, HF\_vert calculated in BEP and PALM scales with  $-\frac{\eta \Delta T}{c_p}$  where  $\eta$  is

a O(10) constant,  $c_p$  is the specific heat O(106), and  $\Delta T$  is the potential temperature

difference between the air and wall. As a result, HF vert is on the order of  $10^{-5}\Delta T$ .

At the roof, HF hor in BEP and PALM follow the M-O parameterization, where HF hor scales

with
$$-\left[\frac{k}{\log\left(\frac{z}{z_0}\right)}\right]^2 U\Delta T$$
, where  $k=0.4$ ,  $z$  is  $\frac{1}{2}\Delta z_1$ ,  $U$  the wind speed. Eventually, HF\_hor is on

the order of  $O(10^{-3})\Delta T$ . Clearly, the heat flux is more negative at the roof compared to that within the canyon. Therefore, the drastic reduction of heat flux was observed at the roof level.

The revised texts are shown in lines 262 to 264.

1.349 '....to a minimum (maximum magnitude) value'. Please carefully review the use of maximum/minimum and decrease/increase in the entire paragraph (given the fact that the momentum fluxes are negative).

We have revised the wordings such that the magnitude of momentum fluxes was discussed in lines 274 to 277.

1.361 The two PBL schemes...

Revised accordingly.

1.400 Figures C1 and C2 in Appendix C.....

We have revised the texts so that readers are referred to Appendix C.

Fig. 9, caption: ,.... for the grid point of the observational sites (Fig. 2) USTSS (LCZ5), HAT (rural), and KP (LCZ1) from ... '

Revised accordingly.

1.434 In fact, L is most commonly referred to as 'Obukhov length' (Obukhov has published the length scale already some six years before the joint paper with Monin in 1954. Should be corrected in the entire manuscript.

"Monin-Obukhov length" has been revised to "Obukhov length" throughout the manuscript.

Fig. 13, caption: again 'reference' instead of 'ground truth'

Revised accordingly.

1. 498 as 1. 400

We have revised the texts so that readers are referred to Appendix C.

1.523 'remained positively skewed'. I don't think we see the result of a skewed distribution here. They remain positively biased.

The word "skewed" has been replaced by "biased".

Fig. 20 why not the same colors as in Figs. 6 and 13?

Figure 20 is now using the same color scheme as Figs.6 and 13 for consistency.

1. 535 ....were considerably smaller...'

Revised accordingly.

1.571 TKE-ACM2+Bulk

Revised accordingly.

"Coupling the TKE-ACM2 Planetary Boundary Layer Scheme with the Building Effect Parameterization Model" by Wanliang Zhang, Chao Ren, Edward Yan Yung Ng, Michael Mau Fung Wong, and Jimmy Chi Hung Fung

Recommendation: Minor revisions

**General comments**

This manuscript introduced an approach coupling the recently developed TKE-ACM2 PBL scheme with the multi-layer BEP UCM in the WRF model (hereafter, TKE-ACM2+BEP) and evaluated the performance of the TKE-ACM2+BEP approach in comparison to idealized LES for two convective PBL (CBL) cases as well as 1-month real-case observations. For both idealized and real cases, the simulations using the TKE-ACM2+BEP approach were compared with the TKE-ACM2+Bulk to investigate impacts of UCM (BEP vs. Bulk), with the Boulac+BEP to investigate impacts of PBL, as well as with Boulac+Bulk that differs in both PBL and UCM.

The coupling between the TKE-ACM2 PBL scheme and the BEP UCM scheme was made by 1) adding the forcing term computed by the BEP UCM to the rhs of TKE-ACM2 PBL tendency terms for prognostic variables including TKE, and 2) modifying the length scale in the TKE-ACM2 closure by considering the length scale comparable to the building height (1 build). The TKE-ACM2+BEP was verified in comparison to idealized LES for two CBL cases focusing on mean temperature and wind profiles and their corresponding vertical flux profiles. The real-case verifications in comparison to lidar wind profiles and near-surface meteorological parameters (U10, T2, RH2) were made for different land use categories, including 10 LCZ urban categories, water, and non-urban categories.

This manuscript is well written and organized providing details of the implementation that are essential to understand the coupling approach and verification results confirming that TKE-ACM2+BEP was properly implemented, showing great potential of TKE-ACM2+BEP to improve the WRF simulations for urban PBLs. I have several minor comments and

suggestions.

We would like to express our sincere gratitude for the time and effort you dedicated to reviewing our manuscript. In response to your suggestions and comments, we have made several improvements to the manuscript.

Minor comments

Lines 114–115, "Ai and Bi": Could you mention here that Ai and Bi are outputs from the BEP UCM? You mentioned it later at Line 125 for TKE, but I suggesting mentioning it to here as well.

We have added extra texts to remind readers that Ai and Bi are outputs from BEP.

Line 146, "anisotropic": I think in general turbulence closure models in LES compute impacts of isotropic turbulence (assuming model resolution is in the inertial subrange), not anisotropic turbulence. Could you double check this for the PALM model? Indeed, the turbulence closure model used in the PALM setting is isotropic according to Equation 13 in Maronga et al. (2015).

.

Line 254, "observations": I guess you mean "LES".

Revised accordingly.

Lines 293–294, "Boulac+BEP seemed to largely underestimate the momentum flux": Due to the underestimation of the momentum flux, I expected Boulac+BEP would overestimate u/ug compared to TKE-ACM+BEP, like TKE-ACM2+Bulk and Boulac+Bulk underestimate the momentum flux and overestimate u/ug, for both 10WC and 24SC cases. However, Boulac+BEP rather underestimates winds compared to TKE-ACM+BEP. Could you explain why the momentum flux and the wind profiles aren't consistent for Boulac+BEP? It shows momentum flux profiles closer to TKE-ACM2+Bulk and Boulac+Bulk, but u/ug profiles closer to TKE-ACM2+BEP.

Not only the magnitude of  $\overline{w'u'}$  affects the  $u/u_g$  profile, but also the gradient of  $\overline{w'u'}$   $(\partial \overline{w'u'}/\partial z)$  is critical because the tendency term  $(\partial u/\partial t)$  is balanced by  $-\partial \overline{w'u'}/\partial z$  as shown in Eqn.1.

$$\frac{\partial u}{\partial t} = -\frac{\partial \overline{w'u'}}{\partial z}$$
 Eqn.1

Below z/H=1, Boulac+BEP produced significantly larger absolute value of  $\partial \overline{w'u'}/\partial z$  than Boulac+Bulk and relatively smaller  $|\partial \overline{w'u'}/\partial z|$  than TKE-ACM2+BEP, consistent with the considerably smaller u than Boulac+Bulk and slightly larger u than TKE-ACM2+BEP. Yet, the relationship between momentum flux and wind speed is not always linear as  $\overline{w'u'}$  is computed mainly by the eddy viscosity times  $\partial u/\partial z$ , and the feedback mechanisms in the boundary layer can influence how wind profiles respond to changes in momentum flux.

Lines 414–415, "The four simulations generated T2 diurnal cycles with much lower amplitude than observations at water surfaces": Could you confirm if the observations were also made over water surfaces (e.g., buoy) or they are over land surfaces surrounded by oceans? The observations show quite strong diurnal cycles of T2, which is not typical over water surfaces. If the observations were made over land surfaces while the simulations were at ocean grid points, this could be the reason why the T2 diurnal cycle amplitude is much lower in the simulations.

A few stations were identified as located on a water grid point in simulations at the horizontal resolution of  $\Delta x = \Delta y = 1$  km, yet they are in fact situated on land. For example, the Green Island station (GI\_AWS, Fig.S46 in the supplement) is placed on a small island surrounded by ocean, leading to that WRF recognizes the grid point as water surface. Other mismatch is found at SHL\_AWS (Fig.S59) and SHW\_AWS (Fig.S60) stations which are placed at the coast. The mismatch between the actual land cover type of the station location and the model landuse is rather common (e.g., Ribeiro et al. (2021)) and inevitable, considering that we needed to manually check the identified landuse of the stations when the simulation resolution and grid setting change.

Extra explanation has been added to line 420 in the revised manuscript.

**References**

- Maronga, B., Gryschka, M., Heinze, R., Hoffmann, F., Kanani-Sühring, F., Keck, M., Ketelsen,
  K., Letzel, M. O., Sühring, M., & Raasch, S. (2015). The Parallelized Large-Eddy
  Simulation Model (PALM) version 4.0 for atmospheric and oceanic flows: Model
  formulation, recent developments, and future perspectives. *Geoscientific Model Development*, 8(8), 2515–2551. https://doi.org/10.5194/gmd-8-2515-2015
- Ribeiro, I., Martilli, A., Falls, M., Zonato, A., & Villalba, G. (2021). Highly resolved WRF-BEP/BEM simulations over Barcelona urban area with LCZ. *Atmospheric Research*, 248, 105220. https://doi.org/10.1016/j.atmosres.2020.105220

---

## Author Response (AR3)

Reviewer 1 – Heat flux profiles (Line 333)

1) A more systematic explanation of the heat-flux profiles is needed in relation to Eq. (16) and the framework of Martilli et al. (2002). The newly added sentence appears to restate the preceding point (reduced heat flux near the roof level) without clarifying the underlying mechanism. Please make the causal chain explicit—i.e., how the terms in Eq. (16) lead to the observed profile features around the roof layer. If possible, please use the full and exact equations used in heat flux calculations of each parameterizations.

We sincerely appreciate your expertise in highlighting the questions regarding our manuscript. In the revised version, we have referenced and cited the relevant equations for calculating the heat flux between the horizontal and vertical surfaces and the atmosphere.

The revised texts can be found in below, as well as in lines 269 to 280 in the revised manuscript.

It is important to emphasize that the heat flux forcing caused by urban effects, represented by term (F) in Eqn.3, consists of the heat flux between the vertical surfaces of the building and the air  $(\overline{w'\theta'}_{\text{vert}})$  as well as the heat flux between the horizontal surfaces and the air  $(\overline{w'\theta'}_{\text{hor}})$ , as noted by [Martilli et al., 2002]. A scale analysis based on [Martilli et al., 2002] (see Eqn.10) reveals that  $\overline{w'\theta'}_{\text{vert}}$  is proportional to  $-(\theta-\theta^{\text{wall}})\eta/c_p \approx -\mathcal{O}(10^{-5})\Delta\theta$ , where  $\eta$  is  $\mathcal{O}(10)$  and  $c_p = 10^{-6}\,\text{J/m}^3/\text{K}$ .

$$\overline{w'\theta'}_{i,\text{vert}} = -\frac{\eta}{c_n} \left[ \left( \theta_i - \theta_i^{\text{West wall}} \right) + \left( \theta_i - \theta_i^{\text{East wall}} \right) \right] S_i \tag{10}$$

where the superscript of  $\theta^{\text{wall}}$  indicates the orientation of walls for an exemplary North-South street direction. In contrast, the heat flux at the roof  $(\overline{w'\theta'}_{\text{hor}})$  scales with  $-\frac{k^2}{(\log(0.5\Delta z_i/z_{0I}))^2}U\Delta\theta$  according to Eqn.13, which is approximately  $-\mathcal{O}(10^{-3})\Delta\theta$ .

$$\overline{w'\theta'}_{I,\text{hor}} = -\frac{k^2}{(\log(0.5\Delta z_i/z_{0I}))^2} U_i \Delta\theta f_h S_I$$
(11)

where  $f_h$  is a stability correction factor used in [Louis, 1979]. Therefore,  $-\overline{w'\theta'}_{hor}$  is by orders of magnitudes greater than  $-\overline{w'\theta'}_{vert}$  at the roof level, the resultant heat flux observed a significant decrease of at z/H = 1.

2) The manuscript suggests that heat conduction is the primary source of the warm bias in bulk models. However, it is not clear whether conduction is actually parameterized in the present configuration. Please clarify which processes are represented (e.g., slab heat storage, conductive transfer through roof/wall/ground layers) and how they enter Eq. (16). If conduction is not parameterized, the explanation should be revised accordingly.

In the revised manuscript, we clarified that different mechanisms dominate heat exchange between different surfaces and air.

Between vertical surfaces (wall) and the air, the heat exchange is modeled as fully conductive, which is shown in Eqn.10 in the revised manuscript using the temperature difference  $\Delta\theta$  and thermal diffusivity.

In contrast, the heat exchange between the horizontal surface and the air is dominated by forced convection as modeled using the bulk aerodynamic method suggested by [Louis, 1979] (see Eqn.11 in the revised manuscript).

However, it should be distinguished that the above-mentioned conduction is different from that which happens at the interior of the materials of buildings. This conductive transfer within the layers of materials is solved using the diffusion equation of heat,

$$\frac{\partial T_i}{\partial t} = \frac{\partial}{\partial z} \left( K_s \frac{\partial T_i}{\partial z} \right) \tag{12}$$

where  $K_s$  is the substrate thermal conductivity of the material. At the boundary, an energy budget equation is supplemented,

$$\frac{\partial T_n}{\partial t} = \frac{\left(HF - K_s \frac{\partial T}{\partial z}\big|_{n-1}\right)}{\Delta z g_n} \tag{13}$$

where  $\Delta z g_n$  is the thickness of the material layer, and HF is:

$$HF = \frac{(1-\alpha)R_s + \varepsilon R_l - \varepsilon \sigma T_n^4 + H}{C_s}$$
(14)

where  $\alpha$  is the albedo of the surface,  $R_s$  is the direct and reflected solar radiation received by the surface,  $\varepsilon$  is the emissivity of the surface,  $R_l$  is the long wave radiation received by the surface, H is the sensible heat flux, and  $C_s$  is the specific heat of the material. The representation of Eqns.12, 13, and 14 are identical in WRF simulations (Bulk and BEP), as well as in PALM LES.

The 'conductive heat transfer' used throughout the manuscript refers to the direct contact of the wall and atmosphere represented by Eqn.10 and is not related to Eqns.12, 13, or 14...

3) This subsection is currently difficult to follow; a concise rewrite that connects equations, physical interpretation, and figures would greatly improve clarity.

We have revised the structure of the manuscript by incorporating additional subsections to clearly distinguish the different parts of the results being discussed. For instance, we added subsections:

ullet 3.1 Turbulence characteristics and runtime parameters

- 3.2 Case 10WC results: vertical profiles of  $\theta$ , u,  $\overline{w'\theta'}$ , and  $\overline{w'u'}$
- 3.3 Case 24SC results: vertical profiles of  $\theta$ , u,  $\overline{w'\theta'}$ , and  $\overline{w'u'}$

to guide readers.

We have also enhanced coherence by referencing equations and figures when needed, e.g., lines 267 to 283. Finally, we improved the clarity of the discussion by providing physical interpretations immediately following the sentences that present the results.

4) Please also demonstrate—quantitatively—that the two bulk models produce different magnitudes of sensible heat flux (e.g., by referencing specific panels/figures and providing summary statistics).

Thanks for the suggestion. We have added the following texts to enrich the understanding of the different magnitudes of sensible heat flux produced by the two Bulk models.

The TKE-ACM2+Bulk simulation produced a heat flux of  $0.124\,\mathrm{K\,m\,s^{-1}}$  ( $0.207\,\mathrm{K\,m\,s^{-1}}$ ) at the ground in Case 10WC (Case 24SC) according to Fig.5a (Fig.5c), which is approximately 1.17 (0.817) times the magnitudes simulated by Boulac+Bulk. Reviewer 2 – Momentum flux profiles (Line 333)

1) The authors' response to the previous comment remains unclear and partially redundant. Please provide a streamlined, explicit explanation of the momentum-flux behavior and incorporate the key points into the main text at Line 333 (rather than only in the response letter).

We have revised the explanation for the momentum flux behavior with the addition of the following texts in lines 314 to 324:

The two Bulk methods produced monotonically increasing momentum flux from the ground to the top of the boundary layer. The  $\overline{w'u'}$  in TKE-ACM2+Bulk was less negative than that in Boulac+Bulk (Fig.5b and Fig.5d), corroborating a relatively larger  $u/u_g$  (Fig.4b and Fig.4d) in both cases. In contrast, the momentum flux profiles simulated by BEP models had a local minimum value at or above the roof level. Below the roof level, TKE-ACM2+BEP yielded slightly more negative w'u' than Boulac+BEP (Fig.5b and Fig.5d) consistently in the two cases, resulting in a lower wind speed (Fig.4b and Fig.4d). From the roof level to the top of the boundary layer, TKE-ACM2+BEP produced massively larger magnitudes of  $\overline{w'u'}$  due to the addition of the non-local flux, yet the  $u/u_a$ were lower in [18H, 27H] ([16H, 21H]) in Case 10WC (Case 24SC) only compared to Boulac+BEP, indicating an inconsistent correlation between  $u/u_q$  and  $\overline{w'u'}$  within Z=1H to Z=18H (Z=16H). This inconsistency is likely due to the fact that TKE-ACM2+BEP produced a more well-mixed boundary layer, resulting in  $u/u_q$  profiles within the mixed layer that exhibited less variability compared to those in Boulac+BEP, which appeared to have a stronger shear.

Reviewer 2 – Momentum flux profiles (Lines 414–415)

1) Please add more detailed information about the observation stations (e.g., coordinates, elevation, LCZ/surface type, instrument height, averaging period) and include explicit cross-references to the relevant figures in the Supplementary Materials.

The elevation at which the LiDAR units measure the wind speeds is described in line 185. Additionally, we have added descriptive texts to elucidate in detail about the information about the observation stations, including the coordinates, elevation, and LCZ type in lines 189 to 204, which can also be found below.

**Lines 189 to 194:**

The LiDAR unit at the Hong Kong University of Science and Technology Supersite (USTSS\_LCZ5) is located on the east coast of Kowloon Island (22.333 °N, 114.267 °E), where the nearest model grid center falls within LCZ 5 (open mid-rise). The second LiDAR, installed on the southeastern peninsula of Hong Kong Island (Hok Tsui, 22.209 °N, 114.253 °E), is surrounded by natural vegetation and referred to as HT\_rural. Lastly, the LiDAR at King's Park (22.312 °N, 114.170 °E) in downtown Kowloon, where the average building height is 60 m [Kwok et al., 2020], is located within an LCZ 1 model grid (compact high-rise), and designated as KP\_LCZ1.

**Lines 195 to 204:**

The coordinates corresponding to each automated weather station (AWS) are retrieved from the little-r formatted files and outlined in the figures in the supplementary material, e.g., Fig.S6 to Fig.S98. The elevation of each AWS is 10 m above the ground. The landuse type of each AWS and LiDAR unit is identified using the nearest model grid point, which is also outlined in the figures in the supplementary material. All observational data presented in this work is 1-hour averaged.

**References**

[Kwok et al., 2020] Kwok, Y. T., De Munck, C., Schoetter, R., Ren, C., and Lau, K. (2020). Refined dataset to describe the complex urban environment of Hong Kong for urban climate modelling studies at the mesoscale. *Theoretical and Applied Climatology*, 142.

[Louis, 1979] Louis, J.-F. (1979). A parametric model of vertical eddy fluxes in the atmosphere. *Boundary-Layer Meteorology*, 17(2):187–202.

[Martilli et al., 2002] Martilli, A., Clappier, A., and Rotach, M. W. (2002). An Urban Surface Exchange Parameterisation for Mesoscale Models. *Boundary-Layer Meteorology*, 104(2):261–304.

---

## Author Response (AR4)

**Public justification (visible to the public if the article is accepted and published):**

- There is no title for code and data availability, author contribution, competing interests, and acknowledgements.

**We have added the titles for sections:**

- Code and data availability
- Author contribution
- Competing interests
- Acknowledgements
- Please revise captions in the supplement. Do not use the LaTeX style description such as \theta

Revised accordingly in the supplement.